# ROCKETEVAL: EFFICIENT AUTOMATED LLM EVALUATION VIA GRADING CHECKLIST

**Tianjun Wei**[*][†] [12], **Wei Wen**[*] [2], **Ruizhi Qiao**[‡2], **Xing Sun** [2], **Jianghong Ma**[‡ 3]
[1] City University of Hong Kong, [2] Tencent Youtu Lab, [3] Harbin Institute of Technology Shenzhen.
tjwei2-c@my.cityu.edu.hk    {jawnrwen,ruizhiqiao,winfredsun}@tencent.com
majianghong@hit.edu.cn

## ABSTRACT

Evaluating large language models (LLMs) in diverse and challenging scenarios is essential to align them with human preferences. To mitigate the prohibitive costs associated with human evaluations, utilizing a powerful LLM as a judge has emerged as a favored approach. Nevertheless, this methodology encounters several challenges, including substantial expenses, concerns regarding privacy and security, and reproducibility. In this paper, we propose a straightforward, replicable, and accurate automated evaluation method by leveraging a lightweight LLM as the judge, named RocketEval. Initially, we identify that the performance disparity between lightweight and powerful LLMs in evaluation tasks primarily stems from their ability to conduct comprehensive analyses, which is not easily enhanced through techniques such as chain-of-thought reasoning. By reframing the evaluation task as a multi-faceted Q&A using an instance-specific checklist, we demonstrate that the limited judgment accuracy of lightweight LLMs is largely attributes to high uncertainty and positional bias. To address these challenges, we introduce an automated evaluation process grounded in checklist grading, which is designed to accommodate a variety of scenarios and questions. This process encompasses the creation of checklists, the grading of these checklists by lightweight LLMs, and the reweighting of checklist items to align with the supervised annotations. Our experiments carried out on the automated evaluation benchmarks, MT-BENCH and WILDBENCH datasets, reveal that RocketEval, when using *Gemma-2-2B* as the judge, achieves a high correlation (0.965) with human preferences, which is comparable to *GPT-4o*. Moreover, RocketEval provides a cost reduction exceeding 50-fold for large-scale evaluation and comparison scenarios. Our code is available at https://github.com/Joinn99/RocketEval-ICLR.

## 1 INTRODUCTION

**Why is automated LLM evaluation necessary?** In recent years, the progress in large language models (LLMs) has been remarkable (Jiang et al., 2024a; Team et al., 2024; Yang et al., 2024), driven by continuous technological advancements. The rapid emergence of new models and techniques has broadened their applications, encompassing both general-purpose textual and visual LLMs, as well as those fine-tuned for specific tasks in various domains. These LLMs exhibit a range of capabilities and performances across different application scenarios. Therefore, evaluating their capabilities effectively has become crucial for guiding their development. Since most tasks performed by LLMs involve human interaction, human preferences are often considered the gold standard for LLM evaluation (Zheng et al., 2023). Currently, crowd-sourcing platforms like CHATBOT ARENA (Chiang et al., 2024) collect a significant number of human votes to evaluate LLMs. However, this approach relies on extensive and long-term human annotation, which is costly and challenging to reproduce and interpret (Ni et al., 2024). Considering the typical applications of LLM evaluation today, which

---

[*]Equal Contribution.

[†]Work done during internship at Tencent.

[‡]Corresponding Authors: Ruizhi Qiao and Jianghong Ma.

[1]Unless otherwise stated, all names of LLM in this paper refer to their "Instruct" or "Chat" versions.

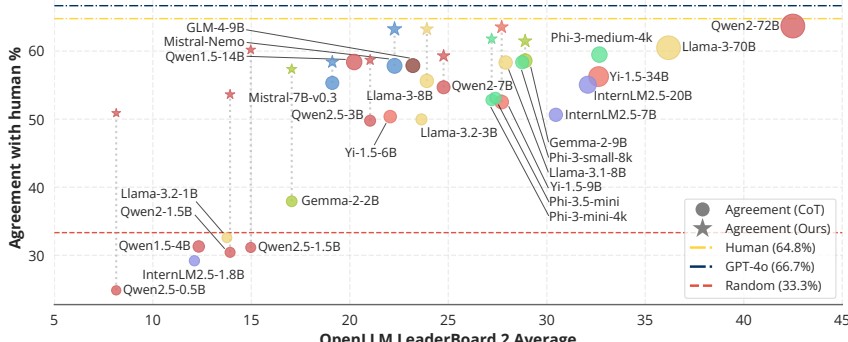

Figure 1: Agreements with MT-BENCH HUMAN JUDGMENTS with different LLM [1]judges. "CoT" indicates the judgments derived using the original Chain-of-thought (CoT) prompting, and "Ours" indicates the judgments derived using our proposed **RocketEval** framework.

include validating the effectiveness of LLM development and assisting users in selecting the best-performing models for specific tasks, there is a growing demand for more efficient, reproducible, and interpretable evaluations of LLMs. This demand has led to the proposal of various methods (Wang et al., 2024b; Kim et al., 2024a) and benchmarks (Lin et al., 2025; Zheng et al., 2023) aimed at achieving reliable and efficient automated LLM evaluation.

**Pros and cons of the existing automated LLM evaluation paths.** Automated LLM evaluation methods can generally be categorized into three primary types:

*Multiple-Choice Questions (MCQ) and Keyword Matching Benchmarks.* These methods evaluate the accuracy of a model's responses by designing a series of closed-ended questions and comparing the model's answers to a predefined standard ground truth. Benchmarks constructed using this approach have proven effective for quickly assessing a model's capabilities across several tasks, such as reasoning (Zellers et al., 2019; Qiu et al., 2020), comprehension (Mihaylov et al., 2018; Liu et al., 2024), and knowledge retention (Hendrycks et al., 2021; Liang et al., 2023). However, the requirement for specific response formats limits the comprehensiveness of this evaluation method. In practice, most LLM applications involve response styles that differ significantly from simple choices and keywords. Notably, many open-ended tasks cannot be judged by a fixed ground truth, which strictly limits the applicability of this approach.

*LLM-as-a-Judge.* Early studies have attempted to adopt language models in automated evaluation (Zhang* et al., 2020; Yuan et al., 2021). Given the robust generalization capabilities of LLMs, employing a powerful LLM as an evaluator has emerged as a viable solution. This method typically involves prompting an LLM to serve as a judge, evaluating the responses from different LLMs to a set of well-designed queries. A crucial prerequisite for LLM-as-a-Judge is that the LLM must possess sufficient capability to fully comprehend the queries and discern the quality of different responses. Consequently, many existing benchmarks (Zheng et al., 2023; Dubois et al., 2023; Li et al., 2024a; Liu et al., 2023) tend to employ the most powerful proprietary LLMs, such as *GPT-4o*, as the judge. However, the use of these models for evaluation not only incurs high costs but also raises other issues such as reproducibility and data privacy.

*Fine-Tuned Judge Models.* These models are designed to address the limitations associated with powerful LLM-as-a-Judge approaches. Fine-tuned judge models, derived from lightweight base models, are trained with high-quality evaluation data to more closely align with human preferences (Jiang et al., 2024b; Zhu et al., 2025; Wang et al., 2024b; Kim et al., 2024a). Compared to proprietary LLMs, such fine-tuned judge models exhibit competitive evaluation capabilities in a more transparent and cost-effective manner. However, simply fine-tuning the model on evaluation task data may degrade other capabilities of the lightweight model, which are already weaker compared to those of powerful proprietary LLMs. This degradation can result in the model failing to correctly understand complex instructions in the queries, thereby deteriorating subsequent evaluation performance (Huang et al., 2024). Additionally, as the capabilities of the base models rapidly evolve and

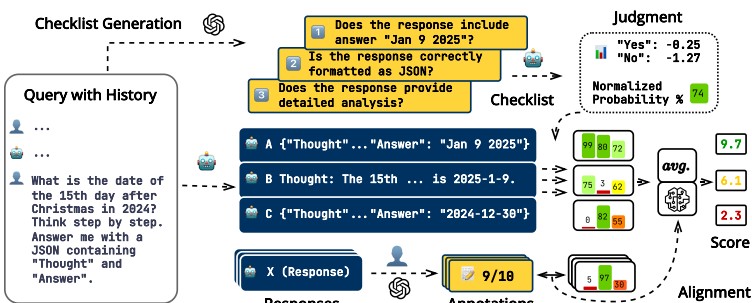

Figure 2: Illustration of **RocketEval** framework for automated LLM evaluation. The framework consists of three components: Checklist Creation, Checklist Grading and Score Prediction.

new data continuously emerges, these judge models may need iterative updates, leading to significant cost escalations and reproducibility issues.

**RocketEval: Towards efficient automated LLM evaluation.** Building on the aforementioned concepts, we find a pathway that synergizes the language modeling and human preference alignment capabilities of the most powerful LLMs with the evaluation efficiency of lightweight models. Specifically, we conduct a thorough analysis of the evaluation capabilities of various large models and observe the following:

1. The agreement between LLM judges and humans is significantly influenced by the inherent capabilities of the LLMs. More powerful LLMs tend to generate evaluation results that more accurately reflect human preferences.

2. Prompt engineering techniques, such as Chain-of-Thought (CoT), exert minimal impact on the evaluation capabilities of the model, particularly when a lightweight LLM is employed as the judge. High uncertainty and positional bias during the decoding process are potential contributing factors to this phenomenon.

Inspired by these observations, we introduce a novel evaluation framework named RocketEval, designed to meet the demands of evaluation scenarios that require high efficiency, low cost, alignment with human preferences, reproducibility, and interpretability. As illustrated in Figure 2, RocketEval operates through a three-stage framework to generate evaluations. Initially, an instance-level checklist is created, providing essential knowledge and critical focus areas to overcome the limitations of lightweight LLMs in constructing analyses. Subsequently, lightweight LLMs assess the quality of responses for each checklist item independently, resulting in a multifaceted and unbiased judgment. The normalized score of each judgment is then collected and aggregated to predict the final score, aiming to mitigate the uncertainty associated with lightweight LLMs. Considering the widespread availability of human annotations across various evaluation scenarios, we further introduce a supervised prediction process to align the scores from lightweight LLMs with these annotations. Experimental results demonstrate that RocketEval significantly enhances agreement with human judgments, achieving a remarkable Spearman correlation of 0.965 when utilizing the *Gemma-2-2B* model as the judge to rank test models. This offers a comparable solution to *GPT-4o* at only 2% of the evaluation cost in large-scale evaluation scenarios, rendering it suitable for performing efficient, reliable, and reproducible LLM evaluations.

## 2 HOW LIGHTWEIGHT LLMS PERFORM AS A JUDGE?

In this section, we verify the capability of lightweight LLMs in automated evaluation by conducting a series of experiments employing various LLM judges.

### 2.1 SETUP

We selected two benchmark datasets for our experiments:

- MT-BENCH (Zheng et al., 2023) is a classic benchmark that includes 80 multi-round queries from multiple domains and uses *GPT-4* as the judge for multi-round evaluations.
- WILDBENCH (Lin et al., 2025) is a newly released benchmark containing 1,024 real-world user queries. WildBench first introduces a manually revised checklist as contextual information to guide the evaluation and uses *GPT-4o* as the judge.

Both benchmarks include pairwise comparison and point-wise scoring evaluation methods. Since the results of pairwise comparisons can be derived from comparing the scores derived from point-wise scoring, we focus on the point-wise scoring method. In point-wise setting, both benchmarks prompt the judge to first generate an analysis of the response, followed by a score in 1-10 as the final judgment. This can be seen as a chain-of-thought (CoT) (Wei et al., 2022) process, aimed at enhancing the ability of the LLM on the evaluation task that involves a reasoning process.

## 2.2 HOW DO LIGHTWEIGHT LLM JUDGES PERFORM?

First, we aim to understand the capabilities of different models when they serve as judges. A key metric for evaluating LLMs' evaluation capability in aligning user preferences is their agreement with humans. Here, we measure this agreement using MT-BENCH HUMAN JUDGMENTS (Zheng et al., 2023), which provides human-annotated results for MT-BENCH across six test models. Each sample includes a target query, responses from two LLMs, and human-annotated match results (including ties). We structure the scores obtained in a point-wise manner into the same format by comparing the scores of each pair of responses. Unlike the previous study (Kim et al., 2024a) that uses sampling decoding to derive results without ties, we follow the setting of Lin et al. (2025) and retain cases with scores difference smaller than 0.1 as ties. We then compared the agreement of different LLM judges with human annotations. The human-human agreement reported by Zheng et al. (2023) and the agreement between *GPT-4o* and human are listed as the baseline. As illustrated in Figure 1, agreement between LLMs and humans typically ranges from human-to-human agreement (64.8%) to lower than random outcomes (33.3%). This suggests that the existing evaluation method using CoT scoring imposes significant demands on the judges' abilities. The ability of LLMs, reflected by OPENLLM 2 (Fourrier et al., 2024) scores, significantly impacts its alignment with human preferences during evaluation, thereby complicating the process of performing efficient and reliable evaluations with lightweight LLMs.

## 2.3 WHERE DO LIGHTWEIGHT LLMS UNDERPERFORM AS A JUDGE?

In this part, our objective is to delve deeper and identify the key components that affect the judgment of lightweight LLMs when they serve as judges. We start by conducting an analysis of the two processes involved in evaluation: analysis generation and scoring. To determine whether a comprehensive analysis can boost or degrade the performance of judgment, we instruct the LLM judges to score the responses under three different settings:

- CoT: The original Chain-of-Thought style, generating the analysis and score step-by-step.
- Direct: The judge is prompted to skip the analysis step and score the response directly.
- CoT$_{\text{GPT-4o}}$: We extract the analysis generated by *GPT-4o* to replace the part in judge model's output, then prompt the judge model to score the response accordingly.

Then, the scores of all responses are averaged to derive the score of the tested model for ranking. For baselines, we derive the score predicted by *GPT-4o* and *Claude-3.5-Sonnet* as judges equipped with CoT. We use the CHATBOT ARENA ELO RATINGS (Hard-prompt English) (Chiang et al., 2024) as the ground-truth rankings of human preferences in these models. Figure 3 shows the scores of 12 test models and the Spearman correlation coefficient of ranking lists with *GPT-4o* when different LLMs served as judge. Also, we convert the scores into pairwise comparison results and compare the agreement of different LLM judges with strong baselines. The result is shown in Table 1. Compared to direct scoring, the process of lightweight models conducting their own analysis did not yield significant gains. However, utilizing analysis from powerful LLM as the context significantly improves the evaluation performance in terms of instance-level agreement and list-level correlations. This indicates that lightweight judges are capable of calibrating their judgments with

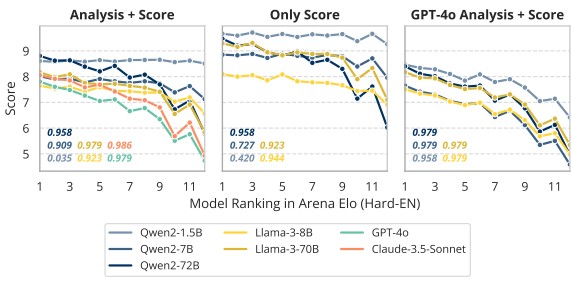

Figure 3: WILDBENCH scores predicted by different LLM judges and the ranking correlation with *GPT-4o*.

Table 1: Agreements of different judges on WILDBENCH.

| | Agreement with GPT-4o | | |
|---|---|---|---|
| | Direct | CoT | CoT$_{\text{GPT-4o}}$ |
| Claude-3.5-Sonnet | | 60.7% | |
| Qwen2-1.5B | **40.1%** | 36.4% | 70.3% |
| Qwen2-7B | 43.1% | **45.0%** | 76.6% |
| Qwen2-72B | 61.0% | **61.4%** | 74.0% |
| Llama-3-8B | 46.7% | **48.4%** | 74.0% |
| Llama-3-70B | 57.2% | **59.3%** | 74.9% |
| | Agreement with Claude-3.5-Sonnet | | |
| GPT-4o | | 60.7% | |
| Qwen2-1.5B | **40.6%** | 36.4% | 58.2% |
| Qwen2-7B | 42.8% | **45.3%** | 60.3% |
| Qwen2-72B | 61.5% | **62.3%** | 62.9% |
| Llama-3-8B | 48.2% | **49.2%** | 61.0% |
| Llama-3-70B | 58.3% | **62.0%** | 63.4% |

powerful LLM when high-quality and comprehensive analysis is provided. In other words, the ability of such lightweight LLM judges is limited mostly due to the poor comprehension and analysis process when they dealing with hard queries and complicated responses.

## 2.4 WHY DO LIGHTWEIGHT LLMS NOT GOOD AT JUDGING?

In the evaluation scenario, conducting analysis can be viewed as a series of judgments on the target response. In order to explore the reasons why the lightweight model are struggling in the conducting analysis, we further conduct a fine-grained level experiments. Specifically, we transform the process of analyzing into a process of judging on a series of checklist questions. We treat each item in the checklist provided in WILDBENCH as a independent question, and prompt the LLM judge to make decisions. Since these questions can be viewed as binary choice questions (for example, *"Does the response correctly identify..."*), the judges are asked to simply output "Yes" or "No" for each question based on the content of the query and response. Then, we obtain judgments of different LLM judges on the checklist questions through repeated sampling, and calculate the ratio of disagreement in different sampling results among all checklist questions. Figure 4 shows that the ratio of disagreement varies significantly across different model sizes. Lightweight LLM such as *Qwen-2-1.5B*, shows an disagreement ratio exceeding 50% across 3 sampling results. This indicates a high uncertainty in making decisions on these checklist questions.

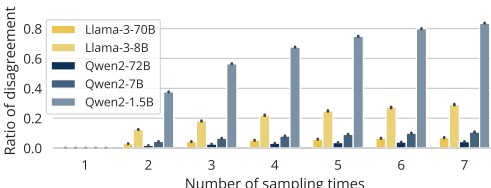

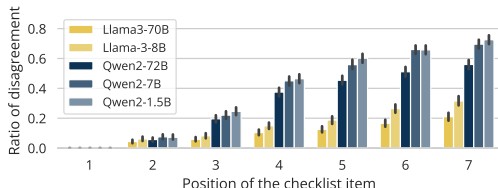

Figure 4: Ratio of disagreement on WILDBENCH with different number of sampling times.

Figure 5: Ratio of disagreement on WILDBENCH with checklist items in different positions.

When lightweight LLM judges conduct the CoT style analysis, this high uncertainty may lead to greater deviations in the final scoring results and degrade the performance. Although previous studies (Li et al., 2024b; Wang et al., 2024a) has demonstrated bias of response order in pairwise comparisons, bias in the analytical process remains unexplored. Therefore, we want to understand whether this form can affect the models' judgment results. We transform the item-level questions from the previous step into a multi-turn dialogue format, with the sequence consistent with the original checklist order. To show the impact of different judgment results, we set all previous judgments to "Yes" or "No" and then compare the disagreements in the current item's judgment results under the two settings. As shown in Figure 5, with an increase in the number of previous questions, the proportion of inconsistencies in all models shows a growing trend. Interestingly, the *Llama-3* series show an overall higher consistency compared to the *Qwen-2* series. Additionally, we notice a correlation

between model size and consistency, with smaller models tending to produce higher inconsistency. This suggests that the uncertainty of lightweight LLMs is more likely to be amplified in the process of setting up sequential analyses with CoT, thus reducing confidence in automated evaluation.

From the above analysis, we demonstrate that lightweight LLM judges exhibit high uncertainty and position bias, which can lead to difficulties in making reliable judgments. A feasible and efficient evaluation method should avoid the above defects, which inspired us to propose the **RocketEval**.

## 3    METHODOLOGY

In this section, based on the analysis on existing evaluation benchmarks, we introduce a new automated LLM evaluation framework named **RocketEval**. As shown in Figure 2, the entire **RocketEval** framework can be divided into three stages. First, we employ a powerful LLM such as *GPT-4o* to conduct meta-analysis and create a checklist for assessing user query. Subsequently, lightweight LLMs are employed to evaluate the checklist items for responses from each test model. Finally, the evaluations for each item are collected to derive the final score, which can be predicted using either an unsupervised arithmetic mean or a supervised predictor learned from annotations.

### 3.1    CHECKLIST CREATION

Evaluating LLM responses is challenging for both human evaluators and LLM judges. Human evaluations can be subjective, while LLM judges may struggle with query understanding, detailed analysis, and interpretation, especially when the lightweight LLMs are employed as the judge. Existing methods improve the accuracy of LLM judgments by providing additional information to the LLM Judge's analytical process, such as reference answers (Zheng et al., 2023; Dubois et al., 2023) and hand-crafted rubrics (Kim et al., 2024a). However, reference answers are of limited use in open-ended queries, while hand-crafted rubrics face a trade-off between labor cost, generalizability, and accuracy. Therefore, here we follow WILDBENCH (Lin et al., 2025) to create a instance-specific checklist, guiding judge LLMs in evaluation. The checklist items are expected to have the following characteristics: 1) Relevance to the topic of the query. 2) Capability to effectively distinguish between different responses. 3) Independence from each other, as independent questions can ensure a complete evaluation of the response's quality. Essentially, checklist questions can be considered as a distillation of knowledge from powerful LLMs to prompt the lightweight judges. A question like *"Does the response include the correct reasoning steps/final answer as X?"* can be helpful when lightweight LLMs are struggling to identify all key factors or solve the problem by itself. We employ *GPT-4o* as the checklist creator, with 5-10 questions created for each instance. This process only needs to be executed once, while the checklist can be leveraged by lightweight LLM judge to evaluate any number of responses. The prompts used are shown in Appendix A.1.

### 3.2    CHECKLIST GRADING

In Section 2.3, we examine the limitations of employing lightweight LLMs as judges, particularly the issues of high uncertainty and positional bias. To address these challenges, we propose the following two evaluation procedures when using lightweight LLMs as the judge.

**Independent Checklist Item Judgment.** To avoid interference from the judgments on other checklist questions, we prompt the LLM judge to evaluate each question in the checklist independently. While this method requires multiple queries per instance, the computational cost can be significantly reduced by leveraging prefix caching (Zheng et al., 2024) since they share the same prefix.

**Normalized Score.** Given the high uncertainty associated with lightweight LLMs, relying solely on binary outcomes such as "Yes" or "No" can result in significant errors in the final judgment. To mitigate this error, we introduce the conditional normalized score as the basis for judgment, which accounts for the certainty of the result. Assuming the probability of output token $t$ from judge LLM parameterized by $\boldsymbol{\theta}$ with context $x$ is $p_{\boldsymbol{\theta}}(t|x)$, with the target query with context $\boldsymbol{x}$ and the corresponding response $\boldsymbol{y}$. Then the conditional normalized score on checklist item $\boldsymbol{c}$ is defined as:

$$\hat{p}(\boldsymbol{x}, \boldsymbol{y}, \boldsymbol{c}) = \frac{p_{\boldsymbol{\theta}}(\text{Yes}|\boldsymbol{x}, \boldsymbol{y}, \boldsymbol{c})}{p_{\boldsymbol{\theta}}(\text{Yes}|\boldsymbol{x}, \boldsymbol{y}, \boldsymbol{c}) + p_{\boldsymbol{\theta}}(\text{No}|\boldsymbol{x}, \boldsymbol{y}, \boldsymbol{c})}. \tag{1}$$

In this manner, the judgments with low certainty have less significant impact on the final judgment.

### 3.3 SCORE PREDICTION

After obtaining all judgments for a single query-response instance, we can predict the final score by the normalized scores of all checklist items. Given the checklist $\boldsymbol{c} = [\boldsymbol{c}_1, \boldsymbol{c}_2, ..., \boldsymbol{c}_N] \in \mathcal{C}$ and the normalized scores $\boldsymbol{p} = [\hat{p}(\boldsymbol{x}, \boldsymbol{y}, \boldsymbol{c}_1), \hat{p}(\boldsymbol{x}, \boldsymbol{y}, \boldsymbol{c}_2), ..., \hat{p}(\boldsymbol{x}, \boldsymbol{y}, \boldsymbol{c}_N)] \in \mathcal{P}$. The score predictor $f : \mathcal{S} \to \mathbb{R}$ can predict the score $s$ by $s = f(\boldsymbol{p})$. This methodology ensures a more reliable and accurate evaluation by addressing the inherent uncertainties and biases in lightweight LLM judgments. The score predictor can be any statistical method, hand-crafted rules, or machine learning models. For simplicity, here we use the arithmetic mean of all normalized scores as the score $s_{unsup}$:

$$s_{unsup} = \sum_{i=1}^{N} \hat{p}(\boldsymbol{x}, \boldsymbol{y}, \boldsymbol{c}_i). \tag{2}$$

This method does not require additional data or effort to obtain the predictor. However, checklist items may have varying impacts on the final results. In many LLM evaluation scenarios, such as LLM development, the benchmark data often comes with annotations from humans and powerful LLMs that serve as the gold standard for evaluation. In this context, we can utilize these annotations to align our checklist judgment results with supervised learning. Specifically, we consider the judgments on the checklist items as features and the annotations as labels. Given a judgment set with $N$ samples, with features $\mathbf{P} \in \mathcal{P}^N$ and labels $\boldsymbol{r} \in \mathbb{R}^N$, a predictor $f_{sup} = min_{\boldsymbol{\theta}} \mathcal{L}(\boldsymbol{\theta}; \mathbf{P}, \boldsymbol{r})$ can be derived by minimizing any loss function $\mathcal{L}$. The predictor can be any classification or regression model, depending on the type of annotations. In this case, we select the Extremely Randomized Tree (Geurts et al., 2006) as estimator to learn a robust ensemble predictor with a limited number of annotations and unknown distributions of the judgment results.

Meanwhile, we have observed that queries may not always yield ideal separable annotations for predictor learning. Some queries may be too easy, too hard, or have vague descriptions, resulting in similar good or bad performance across all test models. This can cause significant performance degradation when learning a supervised predictor. Therefore, we propose a strategy to adjust the impact of the scores predicted by the supervised predictor based on the distribution of the annotations. Specifically, given the annotations $\boldsymbol{r} \in \mathbb{R}^{|\mathcal{P}|}$, we define the weight factor $\alpha_{\boldsymbol{r}}$ as

$$\alpha_{\boldsymbol{r}} = \frac{\epsilon - KL(P_{\boldsymbol{r}} \| P_{ideal})}{\epsilon}, \ \epsilon = \max_{X \sim \mathbb{R}^N} KL(X \| P_{ideal}), \tag{3}$$

where $\epsilon$ is the maximum Kullback–Leibler (KL) divergence of any distribution $X$ from the ideal distribution $P_{ideal}$. In existing work (Kim et al., 2024a; Murugadoss et al., 2024), rubrics have been used as a reference in evaluation, including examples or standards for different rating levels. Therefore, we expect the annotated scores to be varied at different levels, providing the rubrics for the predictor. Hence, we use the uniform distribution across the score range (for example, 1-10) as $P_{ideal} \sim U^N$.

After deriving the weight factor $\alpha_{\boldsymbol{r}}$ and a fitted predictor $f_{sup}$ for each query, the final score assigned to the corresponding response is

$$s_{sup} = (1 - \alpha_{\boldsymbol{r}}) s_{unsup} + \alpha_{\boldsymbol{r}} f_{sup}(\boldsymbol{p}). \tag{4}$$

This methodology ensures a more reliable and accurate evaluation by addressing the inherent uncertainties and biases in the supervised learning process.

## 4 EXPERIMENTS

In this section, we evaluate the effectiveness of the **RocketEval** framework for automated evaluation of LLMs. Initially, we analyze how well **RocketEval** aligns with human preferences at both the instance and the list levels. Next, we investigate the expenses associated with the evaluation procedure. Lastly, we perform an analysis of the checklist's content and its impact on the evaluation.

## 4.1 HUMAN AGREEMENT ON THE EVALUATION

The initial step involves comparing the proposed **RocketEval** framework with the traditional evaluation method, both with and without the inclusion of Chain of Thought (CoT). The idea of adapting checklist or aspect to enhance the robustness in LLM evaluation has been widely adopted in existing works (Lee et al., 2024; Zhou et al., 2024; Fu et al., 2024), where most of them come with a fixed human-curated lists to evaluate responses. To further validate the impact of the instance-level checklist, we introduce a baseline employing six fixed questions as the checklist. These questions, derived by analyzing the MT-BENCH (Zheng et al., 2023) evaluation prompt, encompass dimensions such as helpfulness, relevance, accuracy, depth, creativity, and detail. This baseline is henceforth referred to as the "Fixed". Details of the experimental setup are provided in Appendix A.3.

**Instance-level Agreement.** To measure the agreement ratio of different automated evaluation methods with MT-BENCH HUMAN JUDGMENTS, we adhere to the settings outlined in Section 2.2. As illustrated in Table 2, the proposed method consistently enhances agreement with human judgment across various LLMs acting as evaluators. Notably, the latest 7-12B parameter lightweight LLMs, such as *Llama-3-8B* and *Mistral-Nemo*, achieve over 64% agreement with human judgments and attaining a similar level of agreement as 70B open-source LLMs and human-to-human agreement. For smaller-sized LLMs, which initially exhibit agreement accuracy close to random choice, **RocketEval** significantly improves performance to over 60%, outperforming *GPT-4*. Conversely, the fixed checklist shows no notable performance enhancement and may even degrade performance. This suggests that checklist questions tailored to the specific query topic can better provide a knowledge context, thereby enhancing the performance of lightweight LLMs as judges.

Table 2: Agreement ratios of different LLM judges with MT-BENCH HUMAN JUDGMENTS.

| Method | CoT | Direct | Fixed | Ours (Unsup.) | Ours (Sup.) |
|---|---|---|---|---|---|
| Baseline | GPT-4 (Pairwise): 65.8% Human-to-human: 64.7% Prometheus-7B-v2.0: 55.7% | | | GPT-4 (Single): 59.6% GPT-4o: 66.6% | |
| Llama-3-70B | 60.4% | 64.8% | - | - | - |
| Qwen2-72B | 63.6% | 64.0% | - | - | - |
| Mistral-Nemo | 57.8% | 53.0% | 53.1% | 63.2% | **64.2%** |
| Llama-3-8B | 55.6% | 51.4% | 46.9% | **63.8%** | 62.9% |
| Qwen2-7B | 54.7% | 47.6% | 42.1% | 58.6% | **59.8%** |
| Mistral-7B-v0.3 | 55.3% | 54.4% | 42.8% | **58.8%** | 58.3% |
| Phi-3-mini-4k | 52.8% | 33.8% | 42.1% | **61.2%** | 60.9% |
| Qwen2.5-3B | 49.8% | 52.2% | 35.6% | 57.4% | **58.7%** |
| Llama-3.2-3B | 53.2% | 26.6% | 54.1% | 58.6% | **58.8%** |
| Gemma-2-2B | 37.9% | 39.2% | 43.9% | **57.9%** | 57.3% |
| InternLM2.5-1.8B | 29.2% | 22.3% | 38.4% | **51.7%** | 48.4% |
| Qwen2.5-1.5B | 31.1% | 25.1% | 46.5% | **60.7%** | 60.2% |
| Qwen2-1.5B | 30.4% | 21.7% | 41.0% | **56.2%** | 55.6% |
| Llama-3.2-1B | 32.6% | 22.5% | 36.5% | 33.6% | **41.9%** |
| Qwen2.5-0.5B | 24.9% | 25.1% | 40.8% | **54.3%** | 50.9% |

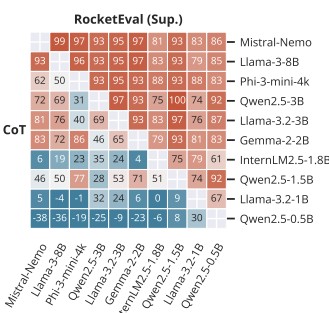

Figure 6: Spearman correlation (in percentage) of test model rankings on WILDBENCH across different LLM judges.

**List-level Correlation.** A critical objective of LLM evaluation is to compare the performance of LLM in specific or general scenarios, which makes the final ranking of LLM essential. Therefore, we conduct the experiment to compare the LLM rankings derived from different approaches. The detailed setup and results are elaborated in Appendix A.3. Table 3 presents the correlation coefficients of the score rankings on WILDBENCH with the CHATBOT ARENA ELO RATING (Hard Prompt - English). The table clearly indicates that **RocketEval** significantly improves the quality of by achieving a higher correlation with human annotations. Specifically, when *Mistral-Nemo* is used as a judge, the Spearman correlation reaches 0.986, surpassing *GPT-4o*. The smaller *Gemma-2-2B* also achieves a correlation of 0.965, surpassing *Qwen2-72B* and the fine-tuned judge model *Prometheus-7B-v2.0* (Kim et al., 2024b). Furthermore, the supervised version of **RocketEval** achieves higher correlations compared to the unsupervised version. Compared to instance-level agreement, the supervised score data provides more significant improvements in list-level correlation. In addition to the agreement with humans, we compare the correlations between different lightweight LLM judges. As shown in Figure 6, **RocketEval** brings significant improvements in cross-judge correlation, especially on the lightweight LLMs. It suggests **RocketEval** exhibits high consistency and reliability in evaluation.

Table 3: Correlation of ranking with CHATBOT ARENA ELO RATING (Hard prompts-English) on WILDBENCH dataset. "Kend." and "Spea." are denoted as Kendall's Tau and Spearman correlation coefficient respectively.

| Method | CoT | | Direct | | Fixed | | Ours (Unsup.) | | Ours (Sup.) | |
|---|---|---|---|---|---|---|---|---|---|---|
| Coefficient | Kend. | Spea. | Kend. | Spea. | Kend. | Spea. | Kend. | Spea. | Kend. | Spea. |
| GPT-4o | 0.909 | 0.979 | - | - | - | - | - | - | - | - |
| Prometheus-7B-v2.0 | 0.848 | 0.949 | - | - | - | - | - | - | - | - |
| Llama-3-70B | 0.909 | 0.979 | 0.787 | 0.923 | - | - | - | - | - | - |
| Qwen2-72B | 0.848 | 0.958 | 0.848 | 0.958 | - | - | - | - | - | - |
| Mistral-Nemo | 0.870 | 0.956 | 0.788 | 0.909 | 0.879 | 0.958 | **0.939** | **0.986** | **0.939** | **0.986** |
| Llama-3-8B | 0.818 | 0.923 | 0.848 | 0.944 | 0.879 | 0.972 | **0.909** | **0.979** | **0.909** | **0.979** |
| Qwen2-7B | 0.788 | 0.909 | 0.545 | 0.727 | 0.636 | 0.804 | 0.758 | 0.895 | **0.818** | **0.930** |
| Mistral-7B-v0.3 | 0.727 | 0.895 | 0.545 | 0.699 | 0.515 | 0.678 | 0.758 | 0.874 | **0.818** | **0.930** |
| Phi-3-mini-4k | 0.424 | 0.587 | 0.576 | 0.762 | 0.182 | 0.273 | 0.788 | 0.916 | **0.848** | **0.958** |
| Qwen2.5-3B | 0.697 | 0.839 | 0.848 | 0.951 | 0.727 | 0.895 | **0.848** | **0.944** | **0.848** | **0.944** |
| Llama-3.2-3B | 0.606 | 0.797 | 0.848 | 0.951 | 0.818 | 0.937 | 0.848 | 0.944 | **0.848** | **0.944** |
| Gemma-2-2B | 0.636 | 0.818 | 0.758 | 0.888 | 0.727 | 0.867 | **0.879** | **0.965** | **0.879** | **0.965** |
| InternLM2.5-1.8B | 0.121 | 0.175 | 0.273 | 0.357 | 0.273 | 0.427 | 0.576 | 0.748 | **0.606** | **0.769** |
| Qwen2.5-1.5B | 0.394 | 0.517 | 0.273 | 0.364 | 0.606 | 0.727 | 0.818 | 0.923 | **0.848** | **0.944** |
| Qwen2-1.5B | -0.061 | 0.035 | 0.303 | 0.420 | -0.061 | -0.014 | 0.455 | 0.622 | **0.667** | **0.825** |
| Llama-3.2-1B | 0.091 | 0.133 | 0.152 | 0.182 | -0.273 | -0.357 | -0.273 | -0.357 | **0.697** | **0.846** |
| Qwen2.5-0.5B | -0.212 | -0.315 | 0.424 | 0.503 | 0.394 | 0.587 | 0.667 | 0.811 | **0.758** | **0.895** |

## 4.2 EVALUATION COST ESTIMATION

In this section, we analyze the costs of various evaluation methods when different LLMs serve as judges. For the purpose of this analysis, we assume that all responses required for evaluation are pre-generated, thereby excluding the inference costs for generating these responses.

In **RocketEval**, the evaluation process is comprised of two primary components: checklist generation and checklist grading. The supervised evaluation method incorporates an additional fitting and prediction process, whose costs are minimal since no LLM inference is involved. For proprietary LLMs serving as judges, we calculate the number of input and output tokens required for a single evaluation and derive the cost based on the official pricing [2]. For open-source LLMs, we deploy them on *NVIDIA RTX A5000* GPUs using *vLLM* (Kwon et al., 2023), and the cost is calculated based on the average execution time and the rental price of the GPU. We reference the pricing on *RunPod* [3] at $0.36 per hour. To maximize efficiency, experiments are conducted in batch mode.

In practical evaluation scenarios, various LLMs, each with distinct tuning options, decoding settings, and prompting techniques, can yield numerous versions of responses. Consequently, evaluations on the same benchmark can be performed hundreds or even thousands of times across different models and their respective versions. We therefore compare the cost of LLM evaluation methods with different number of tests $N$. As shown in Table 4, the cost incurred for generating a checklist for each question is equivalent to the expense of running a single test using *GPT-4o*. Since the checklist generation is a one-time process, the cost of the checklist grading process becomes increasingly significant as the number of tests escalates. For instance, conducting 1000 tests on the WILDBENCH would incur a cost of $3400 when using *GPT-4o* as the evaluator, whereas employing *Llama-3-8B* would only require $71, with achieving a higher correlation with human preferences.

## 4.3 QUALITATIVE ANALYSIS

**Checklist Statistical Analysis.** To discern the distribution of checklist items, we extracted the relationships within the knowledge graphs associated with each checklist item, including the subject, predicate, and object, in a format consistent with the methodology introduced by Sun et al. (2024). We conducted a statistical analysis of all predicates in the checklist items generated from WILD-BENCH (all predicates have been lemmatized). Further statistics are provided in Appendix A.5. As shown in Figure 7, the predicate distribution within checklist items reflects the intrinsic demands of various original questions. Predominant predicates such as *"Include"* and *"Provide"* underscore the necessity for comprehensive and supportive responses, ensuring that all pertinent information is considered. This is crucial for addressing the complex and multifaceted nature of questions across various general tasks. Meanwhile, predicates such as *"Calculate"* and *"Specify"* highlight the pre-

---

[2]https://openai.com/api/pricing/
[3]https://www.runpod.io/pricing

Table 4: Evaluation cost on WILDBENCH with different LLM judges.

| Method | LLM Judge | Deploying Environment | Price | Usage for Single Test | Extra Cost | Total Cost for N Tests | | |
|---|---|---|---|---|---|---|---|---|
| | | | | | | N=10 | N=100 | N=1000 |
| CoT | GPT-4o (20240806) | proprietary | I/O: $1.25 / $5.00 / 1M Tokens | I/O: 1.84M/220k tokens | N/A | $34.0 | $340 | $3400 |
| | GPT-4o-mini (20240718) | | I/O: $0.075 / $0.30 / 1M Tokens | | | $2.00 | $20.0 | $200 |
| RocketEval | Llama-3-70B$_{AWQ}$ | 4 x A5000 | $1.44 / hours | 3760s | $2.87* | $15.1 | $125 | $1224 |
| | Llama-3-8B | 1 x A5000 | $0.36 / hours | 685s | | $3.55 | $9.72 | $71.4 |
| | Gemma-2-2B | 1 x A5000 | $0.36 / hours | 248s | | $3.12 | $5.35 | $27.7 |
| | Qwen2.5-1.5B | 1 x A5000 | $0.36 / hours | 165s | | $3.04 | $4.52 | $19.4 |

*The checklist generation process on WILDBENCH consumes 1.38M input tokens and 228k output tokens on *GPT-4o*.

cision required in quantitative and advisory responses. This distribution pattern not only guides the assessment of answer quality but also ensures that responses meet the specific criteria of each question type, thereby enhancing the overall reliability and applicability of the information conveyed.

**Checklist Item Reweighting.** Adjusting the weights of checklist items is crucial for accurately assessing the effectiveness of responses to the original question, especially when a gold standard is available. Initially, each checklist item was given equal weight, assuming an equal impact on the overall assessment. However, this does not accurately reflect the true importance of each item in validating the response's accuracy and completeness. As shown in Figure 8, Assigning the value of radial stress in item *1* (0.296) is critical, being a core part of the answer, significantly impacts subsequent calculations and the overall analysis. In contrast, item *3* (0.069), which calculates the inner radius, is fundamental but simple, less prone to error, and less critical than stress calculations, thus bearing a lower weight. By reweighting, we emphasize the steps that are the most crucial to the accurate responses, ensuring that the evaluation process is both rigorous and reflective of the actual importance of each component, which leads to more reliable and credible results, ultimately enhancing the overall validity and consistency with the gold standard of the assessment.

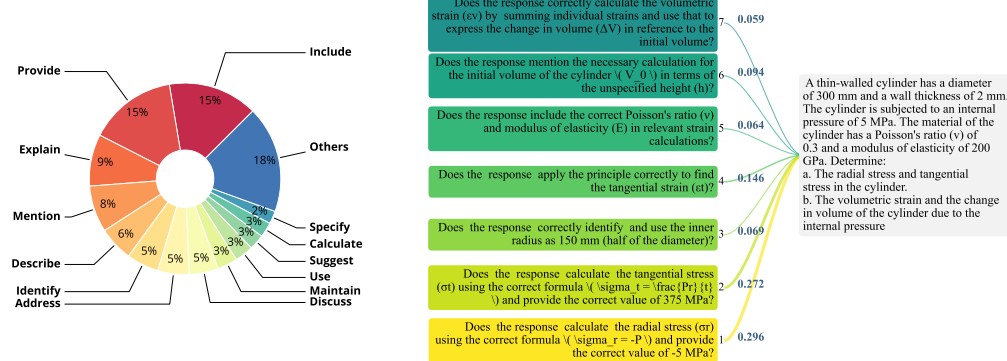

Figure 7: The predicates used in checklist items on WILDBENCH.

Figure 8: Visualization of checklist item reweighting. (Default weight is the reciprocal of the number of checklists)

## 5 CONCLUSION

In this paper, we introduce **RocketEval**, an innovative evaluation framework that uses lightweight LLMs to achieve high efficiency, low cost, interpretability, and alignment with human preferences. By reframing the evaluation task as a multi-faceted Q&A using instance-specific checklists, we addressed the challenges of high uncertainty and positional bias inherent in lightweight LLMs. Our method demonstrated a high correlation with human preferences, achieving a Spearman correlation of 0.965 with the *Gemma-2-2B* model, comparable to *GPT-4o*, but at a fraction of the cost. This significant cost reduction makes RocketEval a feasible solution for large-scale evaluation and comparison scenarios.

ACKNOWLEDGMENTS

This work was partially supported by the National Natural Science Foundation of China (Project No. 62202122 and No. 62073272), the Shenzhen Science and Technology Program under Grant No. GXWD202311301103080001, and the Guangdong Basic and Applied Basic Research Foundation under Grant No. 2024A1515011949.

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

# A APPENDIX

## A.1 PROMPTS USED IN ROCKETEVAL

**Prompts for Checklist Creation.** The following prompt is designed to generate an instance-level checklist.

---

**Prompt for Checklist Creation**

```
# Instruction
You are an helpful assistant who identifies and summarizes key factors in large
    ↪  language models (LLMs) evaluation to help humans evaluate LLMs
    ↪  efficiently.

Feed any query into different LLMs, I will get various responses. I need to
    ↪  know in quick whether these responses follows the instructions and
    ↪  answers the question in the user query better.

I'll provide you with a user query. Your task is to identify those key factors
    ↪  that will affect my judgment and summarize them into a list to improve
    ↪  the efficiency of my evaluation.

# Conversation between User and AI
<|begin_of_history|>

{history}

<|end_of_history|>

## Current User Query
<|begin_of_query|>

{user_query}

<|end_of_query|>

## Reference Response
<|begin_of_reference_response|>

{reference_response}

<|end_of_reference_response|>

# Task
Given the above information, I need you to create a binary question list, so
    ↪  that I can perform an efficient and accurate evaluation through
    ↪  answering several questions.

Your question should be concise and include any necessary key content and
    ↪  information (such as keywords, formats, correct counts and values) in
    ↪  the user query or expected to be shown in responses. Your questions
    ↪  should not only consider evaluating the reference response, but all
    ↪  possible responses. Avoid creating duplicate, cumbersome or vague
    ↪  questions. For example, you should ask "Is this response contain the
    ↪  correct answer ..." instead of "Is this response's answer correct?". Ask
    ↪   fewer questions by aggregating questions with repeated contexts into
    ↪  one question.

## Output Format
Please provide your outputs in the following markdown format by filling in the
    ↪  placeholders in {{}}:
'''
```

---

```
1. {{question1}}
2. {{question2}}
...
‘‘‘
```

**Prompts for Response Judgment.** To keep the consistency with the previous work, we adopt the prompt in Lin et al. (2025) and make several modifications to adapt to different settings.

- For MT-BENCH, to keep consistent with the original paper which utilizes the reference answer to guide the judgment, we replace the part of the checklist with the reference answer and modify the prompt correspondingly.

- For other benchmarks that utilize pairwise comparison, we use the same prompt as WILD-BENCH, and create the checklist using *GPT-4o* for each instance in those benchmarks.

- When prompting LLM to output the score directly without conducting analysis, we modify the output format part in the prompt and change the score range from 1-10 to 0-9. This aims to avoid the ambiguity of scoring 1 or 10 when only the first token is captured.

The prompts with different versions of the modification are listed as follows.

---

**Prompts for Response Judgment**

```
# Instruction
You are an expert evaluator. Your task is to evaluate the quality of the
    ↪ responses generated by AI models.
We will provide you with the user query and an AI-generated responses.
You should first read the user query and the conversation history carefully for
    ↪  analyzing the task, and then evaluate the quality of the responses
    ↪ based on and rules provided below.

# Conversation between User and AI
## History
<|begin_of_history|>
{history}
<|end_of_history|>
## Current User Query
<|begin_of_query|>
{user_query}
<|end_of_query|>
```

**Checklist disabled**

```
## Reference Response
<|begin_of_reference_response|>
{ref_answer}
<|end_of_reference_response|>
```

```
## AI Response
<|begin_of_response|>
{model_output}
<|end_of_response|>

# Evaluation
```

**Checklist disabled**

## Rules
You should first compare the AI response and reference response based on
    ↪ your analysis of the user queries and the conversation history,
    ↪ and then provide your assessment by scoring the AI response.

**Checklist enabled**

## Checklist
<|begin_of_checklist|>
{checklist}
<|end_of_checklist|>
Please use this checklist to guide your evaluation, but do not limit your
    ↪ assessment to the checklist.
## Rules
You should compare the above response based on your analysis of the user
    ↪ queries and the conversation history.
You should first write down your analysis and the checklist that you used
    ↪ for the evaluation, and then provide your assessment according to
    ↪ the checklist.

**CoT enabled**

The scores are in the range of 1~10, where 1 means the response is very
    ↪ poor and 10 means the response is perfect.
Here are more detailed criteria for the scores:

- Score 1~2: The response is very poor and does not make sense at all.
- Score 3~4: The response is poor and does help user solve the problem in
    ↪ a meaningful way.
- Score 5~6: The response is fair but has some issues (e.g., factual
    ↪ errors, hallucinations, missing key information).
- Score 7~8: The response is good enough but could be improved in some
    ↪ ways.
- Score 9~10: The response is perfect and provides helpful information
    ↪ that can help user solve the problem.

## Output Format
First, please output your analysis for the model response, and then
    ↪ summarize your assessment to two aspects: "strengths" and "
    ↪ weaknesses"; Finally, please write down your rating for the
    ↪ assessment.

Please provide your evaluation results in the following json format by
    ↪ filling in the placeholders in []:
```
{
    "strengths": "[analysis for the strengths of the response]",
    "weaknesses": "[analysis for the weaknesses of the response]",
    "score": "[1~10]"
}
```

```
    CoT disabled

    The scores are in the range of 0~9, where 0 means the response is very
        ↪ poor and 9 means the response is perfect.
    Here are more detailed criteria for the scores:

    - Score 0~1: The response is very poor and does not make sense at all.
    - Score 2~3: The response is poor and does help user solve the problem in
        ↪  a meaningful way.
    - Score 4~5: The response is fair but has some issues (e.g., factual
        ↪ errors, hallucinations, missing key information).
    - Score 6~7: The response is good enough but could be improved in some
        ↪ ways.
    - Score 8~9: The response is perfect and provides helpful information
        ↪ that can help user solve the problem.

    ## Output Format
    Please output the score directly as a digit from 0-9. Do not output other
        ↪  text.
    Your score:
```

**Prompts for Checklist Grading.** The prompt for grading checklist items is shown below.

```
    Prompts for Response Judgment

    # Instruction

    You are an expert evaluator. Your task is to evaluate the quality of the
        ↪ responses generated by AI models.
    We will provide you with the user query and an AI-generated responses.
    You should first read the user query and the conversation history carefully for
        ↪  analyzing the task, and then evaluate the quality of the responses by
        ↪ answer the question provided below.

    # Conversation between User and AI

    ## History
    <|begin_of_history|>

    {history}

    <|end_of_history|>

    ## Current User Query
    <|begin_of_query|>

    {user_query}

    <|end_of_query|>

    ## AI Response
    <|begin_of_response|>

    {$model_output}

    <|end_of_response|>

    # Evaluation
```

```
## Question
<|begin_of_question|>

{checklist_item}

<|end_of_question|>

Please answer the given question based on the conversation history and the AI
    ↪ response. You can only answer 'Yes' or 'No'.

Your answer (Yes/No):
```

## A.2 DATASETS AND BASELINES

In addition to the results reported by Zheng et al. (2023) on instance-level agreement and Lin et al. (2025) on list-level correlation, we add more baselines with the following setup:

- *GPT-4o*: We replace *GPT-4* with *GPT-4o* as judge and rerun the experiments using the publicly available code provided by Zheng et al. (2023). For WILDBENCH, we use the judgment of *GPT-4o* provided by Lin et al. (2025) as the baseline results. All baseline results are produced following the template of scoring from WILDBENCH to keep the consistency.

- *Prometheus-2*: We add the state-of-the-art fine-tuned judge model *Prometheus-7B-v2.0* (Kim et al., 2024b) as the baseline. This model introduces custom scoring criteria and rubrics to conduct the evaluation. For simplicity, we use the rubrics in WILDBENCH (Lin et al., 2025), and set the criteria as "The response is in high quality and provides correct, relevant, and helpful information that focuses on the user query.". The complete prompt is shown below.

**Prompts for Prometheus-2 judge**

```
###Task Description:
An instruction (might include an Input inside it), a response to evaluate, a
    ↪ reference answer that gets a score of 5, and a score rubric representing
    ↪  a evaluation criteria are given.
1. Write a detailed feedback that assess the quality of the response strictly
    ↪ based on the given score rubric, not evaluating in general.
2. After writing a feedback, write a score that is an integer between 1 and 5.
    ↪ You should refer to the score rubric.
3. The output format should look as follows: "Feedback: (write a feedback for
    ↪ criteria) [RESULT] (an integer number between 1 and 5)"
4. Please do not generate any other opening, closing, and explanations.

###The instruction to evaluate:
{user_query}

###Response to evaluate:
{model_output}

###Reference Answer (Score 5):
{ref_answer}

###Score Rubrics:
[The response is in high quality and provides correct, relevant, and helpful
    ↪ information that focuses on the user query.]
Score 1: The response is very poor and does not make sense at all.
```

```
Score 2: The response is poor and does help user solve the problem in a
    ↪ meaningful way.
Score 3: The response is fair but has some issues (e.g., factual errors,
    ↪ hallucinations, missing key information).
Score 4: The response is good enough but could be improved in some ways.
Score 5: The response is perfect and provides helpful information that can help
    ↪  user solve the problem.

###Feedback:
```

## A.3 DETAILS OF EXPERIMENTS ON LIST-LEVEL CORRELATION

To comprehensively assess the performance of **RocketEval**, we conduct the experiments by adding two additional benchmark datasets, ALPACAEVAL (Dubois et al., 2023) and ARENA-HARD (Li et al., 2024a). The statistics of all benchmarks are listed in Table 5.

Table 5: Statistics of benchmark datasets.

| Dataset | #Instances | Turns | QueryLen | PromptLen |
|---|---|---|---|---|
| MT-BENCH (Zheng et al., 2023) | 160* | 1-2 | 202.2 | 1123.4 |
| WILDBENCH (Lin et al., 2025) | 1024 | 1-5 | 978.5 | 3402.1 |
| ALPACAEVAL (Dubois et al., 2023) | 805 | 1 | 164.9 | 164.9 |
| ARENA-HARD (Li et al., 2024a) | 500 | 1 | 406.4 | 406.4 |

*Here we treat each 2-turn dialogues as 2 instances.

### A.3.1 EXPERIMENTAL SETUP

We draw inspiration from the study by Lin et al. (2025), where 12 models [4] were selected for a correlation analysis between their rankings in the CHATBOT ARENA ELO RATING (Hard prompts-English) and WILDBENCH. However, upon closer inspection, we notice that some of the chosen models have overlapping elo rating confidence intervals, which may compromise the reliability of the correlation results. To address this issue, we revise the model selection to ensure that there is no overlap in the 95% CI. Similarly, when selecting test models for other benchmarks, we take into account the availability of model responses in the official released data and the elo ratings of the test models. The elo ratings of the selected models are presented in Figure 9.

### A.3.2 RESULTS

**Score Distribution.** We visualize the distribution of the scores from all instances in a single benchmark. Figure 10 shows when lightweight LLMs are employed as the judge and using CoT to grade the responses, the distribution of scores are highly skewed, which indicates the poor ability of such models in distinguish the responses with different qualities. Meanwhile, the scores graded by the same judge under **RocketEval** are highly distinguishable and close to the distribution of *GPT-4o*. Also, the supervised scorer shows the strong ability to reduce the score deviation, making the score distribution closer to the ideal distribution like *GPT-4o*. This suggests the proposed method can provide valuable context information via checklist and guide the LLM to give differential judgment.

**Elo Ratings.** Given that the scores assigned by the same LLM judge can be utilized to establish pairwise comparisons between different test model responses, we follow the approach of (Chiang et al., 2024) by introducing a match simulator and importing all pairwise comparison results from an LLM judge. Specifically, we convert the scores of two responses to determine the winner, with considering differences in scores smaller than 0.1 as ties. For each pair of test models, we derive the

---

[4]The selected models are *gpt-4-turbo-2024-04-09*, *claude-3-opus-20240229*, *Meta-Llama-3-70B-Instruct*, *Qwen1.5-72B-Chat*, *claude-3-sonnet-20240229*, *mistral-large-2402*, *dbrx-instruct@together*, *Mixtral-8x7B-Instruct-v0.1*, *Meta-Llama-3-8B-Instruct*, *tulu-2-dpo-70b*, *Llama-2-70b-chat-hf*, *Llama-2-7b-chat-hf*, *gemma-7b-it* and *gemma-2b-it*.

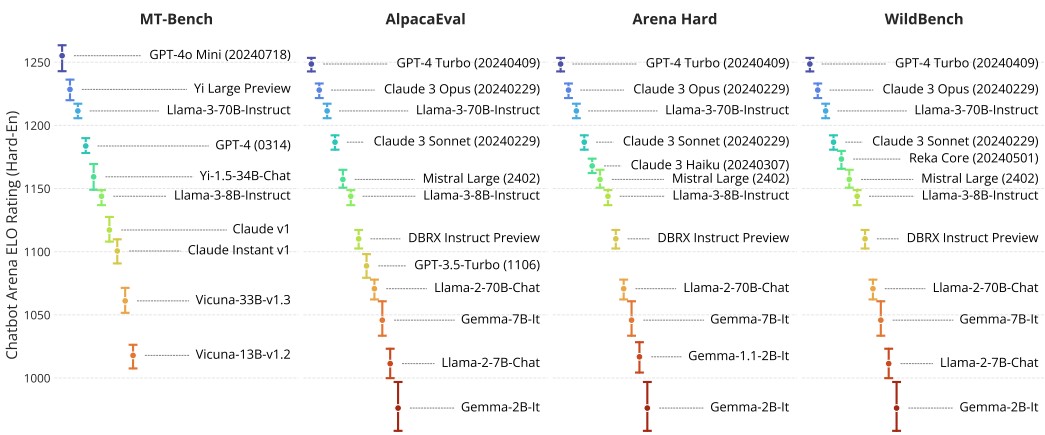

Figure 9: The scores of selected test models on CHATBOT ARENA ELO RATING (Hard prompts English, 2024-09-17).

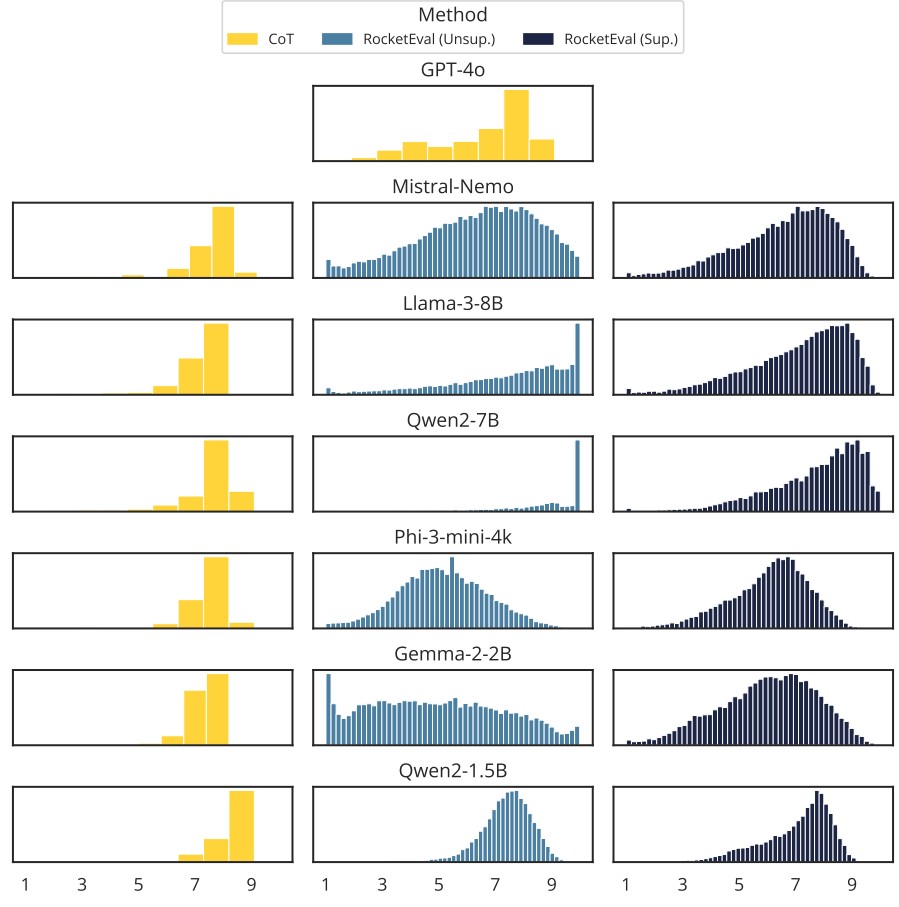

Figure 10: Distribution of scores on WILDBENCH dataset.

pairwise comparison results as the match data. Subsequently, the Elo rating is calculated based on all match results. We employ the same Bradley-Terry model-based Maximum Likelihood Estimation

Table 6: Correlation of ranking with CHATBOT ARENA ELO RATING (Hard prompts-English) on different benchmarks.

**MT-BENCH**

| Method | CoT | | Direct | | Ours (Unsup.) | | Ours (Sup.) | |
|---|---|---|---|---|---|---|---|---|
| Coefficient | Kend. | Spea. | Kend. | Spea. | Kend. | Spea. | Kend. | Spea. |
| GPT-4o | Kendall's Tau (Kend.): 1.0 : Spearman (Spea.): 1.0 | | | | | | | |
| Llama-3-70B | 1.000 | 1.000 | 0.944 | 0.985 | - | - | - | - |
| Qwen2-72B | 1.000 | 1.000 | 0.956 | 0.988 | - | - | - | - |
| Mistral-Nemo | **0.911** | **0.976** | 0.867 | 0.964 | 0.867 | 0.952 | **0.911** | **0.976** |
| Llama-3-8B | 0.867 | 0.952 | 0.867 | 0.964 | 0.822 | 0.927 | **0.911** | **0.976** |
| Qwen2-7B | 0.867 | 0.952 | 0.867 | 0.952 | 0.778 | 0.927 | **0.911** | **0.976** |
| Mistral-7B-v0.3 | 0.733 | 0.903 | **0.822** | 0.927 | **0.822** | **0.939** | 0.822 | 0.939 |
| Phi-3-mini-4k | 0.778 | 0.915 | 0.822 | 0.939 | 0.867 | 0.952 | **0.956** | **0.988** |
| Qwen2.5-3B | 0.764 | 0.912 | 0.644 | 0.867 | **0.822** | **0.927** | **0.822** | **0.927** |
| Llama-3.2-3B | 0.867 | 0.952 | 0.422 | 0.600 | 0.822 | 0.927 | **0.956** | **0.988** |
| Gemma-2-2B | 0.556 | 0.721 | 0.629 | 0.796 | 0.778 | 0.915 | **0.822** | **0.927** |
| InternLM2.5-1.8B | 0.378 | 0.430 | -0.296 | -0.451 | 0.644 | 0.794 | **0.689** | **0.855** |
| Qwen2.5-1.5B | 0.556 | 0.745 | -0.068 | -0.128 | 0.778 | 0.915 | **0.911** | **0.976** |
| Qwen2-1.5B | 0.511 | 0.745 | 0.523 | 0.665 | 0.822 | 0.927 | **0.822** | **0.939** |
| Llama-3.2-1B | 0.511 | 0.697 | 0.114 | 0.146 | -0.067 | -0.200 | **0.733** | **0.879** |
| Qwen2.5-0.5B | 0.333 | 0.394 | 0.422 | 0.479 | 0.600 | 0.733 | **0.600** | **0.806** |

**ALPACAEVAL**

| Method | CoT | | Direct | | Ours (Unsup.) | | Ours (Sup.) | |
|---|---|---|---|---|---|---|---|---|
| Coefficient | Kend. | Spea. | Kend. | Spea. | Kend. | Spea. | Kend. | Spea. |
| GPT-4o | Kendall's Tau (Kend.): 0.879 : Spearman (Spea.): 0.972 | | | | | | | |
| Mistral-Nemo | **0.848** | **0.958** | 0.779 | 0.900 | **0.848** | 0.951 | **0.848** | 0.951 |
| Llama-3-8B | 0.758 | 0.895 | **0.818** | **0.923** | 0.758 | 0.916 | 0.758 | 0.916 |
| Qwen2-7B | 0.667 | 0.825 | 0.606 | 0.734 | 0.788 | 0.923 | **0.818** | **0.937** |
| Mistral-7B-v0.3 | 0.636 | 0.804 | 0.515 | 0.692 | 0.727 | 0.881 | **0.788** | **0.923** |
| Phi-3-mini-4k | 0.606 | 0.762 | 0.565 | 0.708 | 0.818 | 0.937 | **0.818** | **0.944** |
| Qwen2.5-3B | 0.697 | 0.811 | 0.667 | 0.783 | 0.727 | 0.888 | **0.788** | **0.923** |
| Llama-3.2-3B | 0.606 | 0.804 | 0.515 | 0.699 | **0.909** | **0.979** | **0.909** | **0.979** |
| Gemma-2-2B | 0.636 | 0.818 | 0.848 | 0.951 | **0.939** | **0.986** | 0.909 | 0.979 |
| InternLM2.5-1.8B | 0.182 | 0.371 | 0.455 | 0.587 | 0.636 | 0.804 | **0.667** | **0.818** |
| Qwen2.5-1.5B | 0.303 | 0.483 | 0.333 | 0.469 | **0.879** | **0.965** | **0.879** | **0.965** |
| Qwen2-1.5B | 0.273 | 0.420 | 0.303 | 0.406 | 0.545 | 0.706 | **0.818** | **0.909** |
| Llama-3.2-1B | 0.000 | -0.063 | -0.061 | -0.007 | 0.515 | 0.720 | **0.727** | **0.867** |
| Qwen2.5-0.5B | 0.182 | 0.287 | 0.485 | 0.636 | 0.606 | 0.769 | **0.636** | **0.818** |

**ARENA-HARD**

| Method | CoT | | Direct | | Ours (Unsup.) | | Ours (Sup.) | |
|---|---|---|---|---|---|---|---|---|
| Coefficient | Kend. | Spea. | Kend. | Spea. | Kend. | Spea. | Kend. | Spea. |
| GPT-4o | Kendall's Tau (Kend.): 0.939 : Spearman (Spea.): 0.986 | | | | | | | |
| Mistral-Nemo | **1.000** | **1.000** | 0.939 | 0.986 | **1.000** | **1.000** | 0.970 | 0.993 |
| Llama-3-8B | 0.909 | 0.972 | 0.939 | 0.986 | **1.000** | **1.000** | 0.970 | 0.993 |
| Qwen2-7B | 0.901 | 0.974 | 0.879 | 0.965 | **0.970** | **0.993** | **0.970** | **0.993** |
| Mistral-7B-v0.3 | 0.818 | 0.937 | 0.818 | 0.930 | **0.970** | **0.993** | 0.939 | 0.986 |
| Phi-3-mini-4k | 0.879 | 0.958 | 0.758 | 0.881 | **1.000** | **1.000** | **1.000** | **1.000** |
| Qwen2.5-3B | 0.962 | 0.988 | 0.939 | 0.979 | **0.970** | **0.993** | **0.970** | **0.993** |
| Llama-3.2-3B | 0.818 | 0.937 | 0.333 | 0.392 | 0.970 | 0.993 | **1.000** | **1.000** |
| Gemma-2-2B | 0.545 | 0.692 | 0.879 | 0.958 | 0.939 | 0.986 | **0.970** | **0.993** |
| InternLM2.5-1.8B | 0.424 | 0.559 | 0.504 | 0.666 | 0.818 | 0.937 | **0.909** | **0.972** |
| Qwen2.5-1.5B | 0.394 | 0.476 | 0.515 | 0.545 | **0.970** | **0.993** | 0.939 | 0.986 |
| Qwen2-1.5B | 0.121 | 0.224 | 0.636 | 0.748 | 0.879 | 0.965 | **0.939** | **0.979** |
| Llama-3.2-1B | -0.121 | -0.147 | 0.321 | 0.522 | -0.606 | -0.727 | **0.909** | **0.979** |
| Qwen2.5-0.5B | 0.091 | 0.098 | 0.769 | 0.902 | 0.576 | 0.811 | **0.939** | **0.986** |

(MLE), as used by (Chiang et al., 2024), to fit the Elo rating and use bootstrap to estimate confidence intervals.

Figure 11 illustrates the elo rating from CHATBOT ARENA, ARENA-HARD (Li et al., 2024a) and the results of judges *Gemma-2-2B*, *Llama-3.2-3B*, *Qwen2.5-3B*, and the ensemble result on the ARENA-HARD benchmark dataset. We utilize the average score and the elo ratings derived by combining all matches from the three judges. We conduct an experiment on all available responses from Li et al. (2024a) and exclude 10 models used to fit the predictor in **RocketEval** and the *GPT-4o-2024-05-13*, which is used as the judge to produce labels, resulting in 50 test models. It is evident that the scores and elo ratings produced by the LLM judge follow similar trends and are more closely aligned with the results derived from human preferences. Meanwhile, we notice that the test models that exhibit a large deviation from human judgments belong to the same *Llama* series, indicating the potential bias on different patterns of responses.

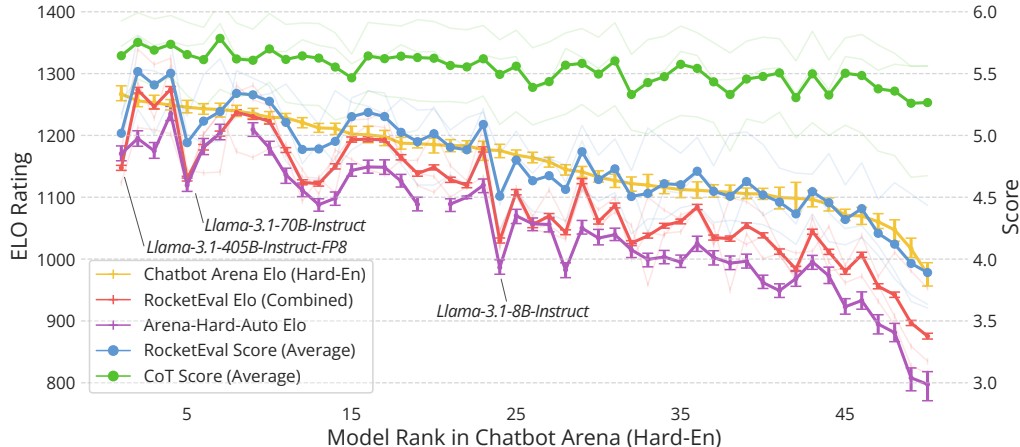

Figure 11: Scores and elo ratings on ARENA-HARD benchmark dataset. We use the CHATBOT ARENA ELO RATING (Hard-En, 2024-09-27) and reproduce the result of ARENA-HARD-AUTO using the official implementation with the default setting (Judge: *GPT-4-1106-preview*, Baseline: *GPT-4-0314*). ARENA-HARD-AUTO Elo, **RocketEval** elo rating and CoT score are adjusted by scaling and adding a constant value for better visualization.

Table 7: Ablation study on instance-level agreement with MT-BENCH HUMAN JUDGMENTS.

| Method | RocketEval (Unsup.) | w/o Norm Score | w/o Indep. Judgment | w/o Weight Factor | RocketEval (Sup.) |
|---|---|---|---|---|---|
| Mistral-Nemo | 63.2% | 62.6% | 63.0% | 63.0% | **64.2%** |
| Llama-3-8B | **63.8%** | 57.5% | 60.8% | 59.9% | 62.9% |
| Qwen2-7B | 58.6% | 49.3% | 59.0% | 57.8% | **59.8%** |
| Mistral-7B-v0.3 | **58.8%** | 47.4% | 57.5% | 52.9% | 58.3% |
| Phi-3-mini-4k | **61.2%** | 55.5% | 60.5% | 57.4% | 60.9% |
| Qwen2.5-3B | 57.4% | 51.3% | **59.2%** | 55.7% | 58.7% |
| Llama-3.2-3B | 58.6% | 48.7% | 57.6% | 56.5% | **58.8%** |
| Gemma-2-2B | **57.9%** | 49.6% | 55.9% | 56.5% | 57.3% |
| InternLM2.5-1.8B | **51.7%** | 39.7% | 42.3% | 37.4% | 48.4% |
| Qwen2.5-1.5B | **60.7%** | 46.3% | 55.7% | 57.5% | 60.2% |
| Qwen2-1.5B | **56.2%** | 31.2% | 54.3% | 52.7% | 55.6% |
| Llama-3.2-1B | 33.6% | 23.4% | 40.9% | **42.0%** | 41.9% |
| Qwen2.5-0.5B | **54.3%** | 33.4% | 49.4% | 47.2% | 50.9% |

## A.4 ABLATION STUDY

To validate the effectiveness of strategies adopted in **RocketEval**, we conduct ablation study by testing the performance on the following variants:

- *w/o Norm Score*: It removes the conditional normalized score and simply use the decoding result as the judgment.
- *w/o Indep. Judgment*: It inputs the checklist into the LLM judge in a multi-turn format, so that the LLM can see its previous judgment result when judging on the current checklist item.
- *w/o Weight Factor*: It sets the weight factor $\alpha_r$ to the constant 1.

The results, presented in Tables 7 and 8, demonstrate that incorporating conditional normalized score consistently enhances the performance of LLM judges, particularly for smaller-sized LLMs. This observation confirms the high uncertainty associated with lightweight LLMs and supports the inference that introducing conditional normalized score can increase their reliability when serving as judges. Simultaneously, setting the weight factor $\alpha_r$ to 1 causes the final score to be entirely determined by the supervised predictor. Predictors trained on a limited number of annotations may struggle to provide accurate scoring results but exhibit superior performance in aligning with human

Table 8: Ablation study on list-level correlation with CHATBOT ARENA ELO RATING (Hard-En) on WILDBENCH dataset.

| Method | RocketEval (Unsup.) | | w/o Norm Score | | w/o Indep. Judgment | | w/o Weight Factor | | RocketEval (Sup.) | |
|---|---|---|---|---|---|---|---|---|---|---|
| Coefficient | Kend. | Spea. | Kend. | Spea. | Kend. | Spea. | Kend. | Spea. | Kend. | Spea. |
| Mistral-Nemo | **0.939** | **0.986** | 0.909 | 0.979 | 0.879 | 0.965 | **0.939** | **0.986** | **0.939** | **0.986** |
| Llama-3-8B | 0.909 | 0.979 | 0.909 | 0.979 | 0.848 | 0.958 | **0.939** | **0.986** | 0.909 | 0.979 |
| Qwen2-7B | 0.758 | 0.895 | 0.758 | 0.895 | 0.788 | 0.916 | **0.909** | **0.979** | 0.818 | 0.930 |
| Mistral-7B-v0.3 | 0.758 | 0.874 | 0.758 | 0.874 | 0.788 | 0.881 | **0.879** | **0.965** | 0.818 | 0.930 |
| Phi-3-mini-4k | 0.788 | 0.916 | 0.758 | 0.902 | 0.818 | 0.930 | **0.909** | **0.979** | 0.848 | 0.958 |
| Qwen2.5-3B | 0.848 | 0.944 | 0.848 | 0.944 | 0.848 | 0.944 | **0.909** | **0.979** | 0.848 | 0.944 |
| Llama-3.2-3B | 0.848 | 0.944 | 0.848 | 0.944 | 0.818 | 0.930 | **0.939** | **0.979** | 0.848 | 0.944 |
| Gemma-2-2B | 0.879 | 0.965 | 0.879 | 0.965 | 0.879 | 0.965 | **0.939** | **0.986** | 0.879 | 0.965 |
| InternLM2.5-1.8B | 0.576 | 0.748 | 0.576 | 0.748 | 0.545 | 0.700 | **0.636** | **0.790** | 0.606 | 0.769 |
| Qwen2.5-1.5B | 0.818 | 0.923 | 0.788 | 0.916 | 0.697 | 0.867 | **0.879** | **0.951** | 0.848 | 0.944 |
| Qwen2-1.5B | 0.455 | 0.622 | 0.515 | 0.643 | 0.636 | 0.804 | **0.758** | **0.874** | 0.667 | 0.825 |
| Llama-3.2-1B | -0.273 | -0.357 | -0.212 | -0.231 | 0.606 | 0.755 | **0.848** | **0.923** | 0.697 | 0.846 |
| Qwen2.5-0.5B | 0.667 | 0.811 | 0.667 | 0.811 | 0.697 | 0.839 | **0.848** | **0.951** | 0.758 | 0.895 |

preferences at the list level. In such scenarios, the weight factor $\alpha_r$ proves to be effective in mitigating the negative influences of biased annotations, thereby achieving strong performance in both instance-level agreement and list-level ranking correlation. Meanwhile, although there is a performance drop in the variant without independent checklist item judgment, the drop is not significant. This may be due to the fact that the position bias exists in all tests and is further alleviated in subsequent score prediction stage. Although the position bias has limited impact on the final prediction results, the form of multi-round dialogue prevents batch processing during LLM inference, thereby reducing efficiency. In conclusion, we believe that independent checklist judgment in **RocketEval** remains an optimal choice.

## A.5 CHECKLIST ANALYSIS

As mentioned in Section 4.3 , we undertake a more detailed examination of the checklists generated by **RocketEval** on the WILDBENCH. This benchmark can be categorized into five major task types: Math & Data Analysis, Coding & Debugging, Creative Tasks, Information/Advice Seeking, and Planning & Reasoning.

In this section, we focus on extracting knowledge graph relationships from all the checklists and conducting a comprehensive analysis of these relationships. Furthermore, we investigate instances of checklist item reweighting across a broader spectrum of tasks to provide a more extensive understanding of the underlying dynamics.

**Subject Distribution.** As shown in Figure 12, the distribution of subject keywords in validating original question responses ensures a universal, compatible, and effective checklist for various tasks. High-frequency keywords like *"explanation"*, and *"code"* are crucial for Coding & Debugging, aiding in verifying code functionality and clarity. Keywords like *"essay"*, *"example"*, and *"story"* are vital for Creative Tasks and Information/Advice seeking, ensuring creativity, relevance, and clarity. For Planning & Reasoning, keywords such as *"strategy"*, *"method"* and *"process"* ensure comprehensive and practical solutions. In Math & Data Analysis, keywords like *"calculation"*, *"algorithm"*, and *"solution"* validate mathematical logic and data analysis. Universally applicable keywords like *"response"*, *"explanation"*, and *"example"* consistently evaluate clarity, relevance, and accuracy across all tasks. This multifaceted approach ensures a robust, flexible, and thorough evaluation process, enhancing the overall effectiveness and reliability of the responses.

**Task-Predicate Relationship.** We categorized the checklist items according to the types of tasks and conducted a statistical analysis of the corresponding predicates. As shown in Figure 13, the distribution of predicate keywords within checklists mirrors the distinct demands inherent to different task categories, such as Coding & Debugging, Planning & Reasoning, Mathematical & Data Analysis, Information/Advice Seeking, and Creative Tasks. Predicate keywords that are exclusive to certain domains, such as *"handle"* within the context of Coding & Debugging or *"calculate"* in Mathematical & Data Analysis, denote actions that are specific and pertinent to those respective

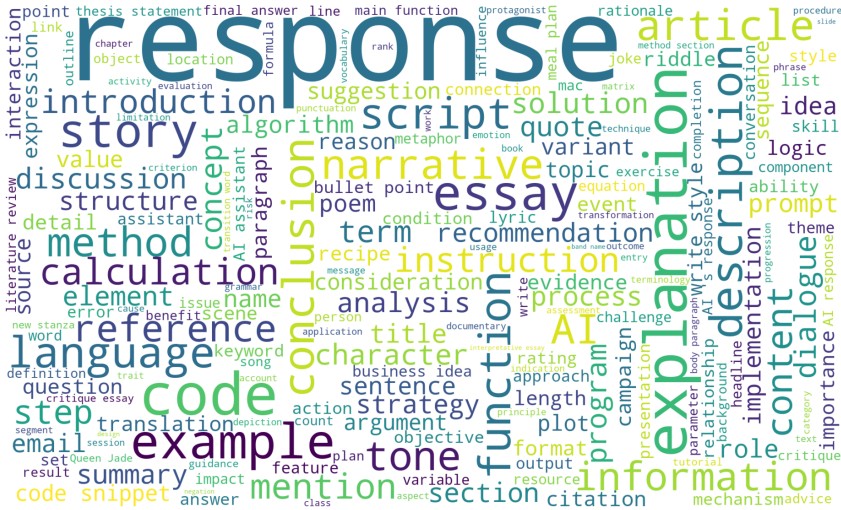

Figure 12: Distribution of Subjects in checklist items which generated from WILDBENCH.

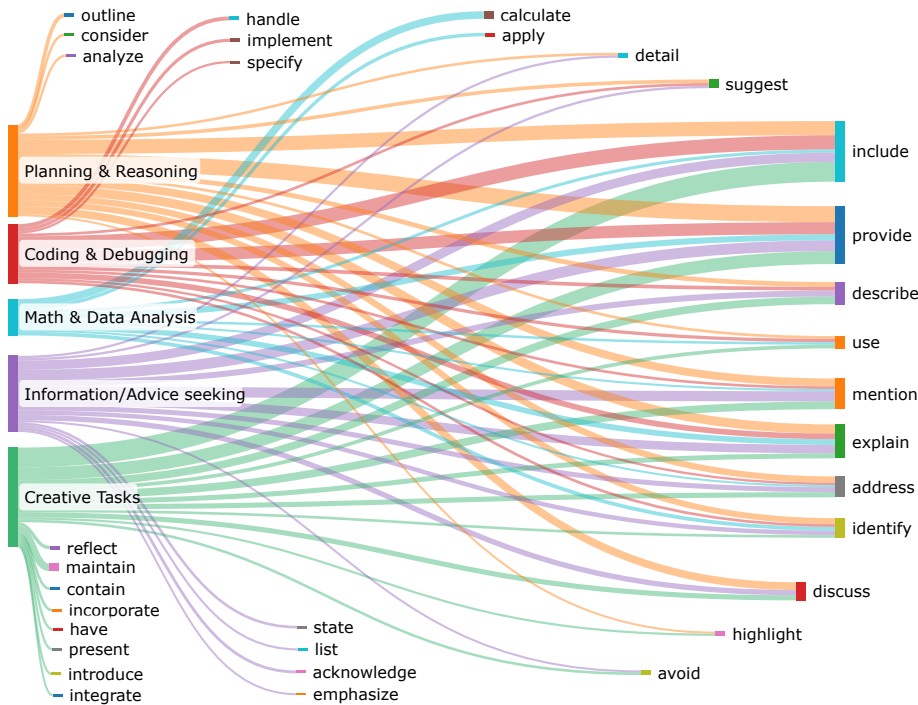

Figure 13: Visualization of Task-Predicate Relationship.

fields. Conversely, common predicates like *"explain"* and *"describe"* are universally applicable, serving general verification objectives across all question types.

These predicate keywords augment the verification process by imparting explicit and goal-oriented directives. Specifically, exclusive keywords concentrate on criteria that are specific to each task type, ensuring a detailed and contextually relevant assessment. Meanwhile, common keywords ensure uniformity and exhaustiveness in the evaluation of responses. This dual strategy ensures

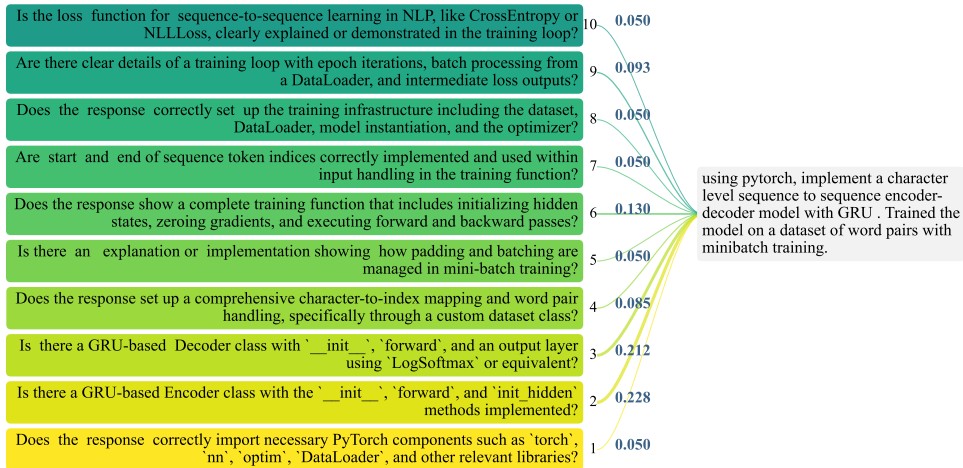

Figure 14: Visualization of checklist item reweighting in Coding & Debugging.

that each answer is evaluated comprehensively and from multiple perspectives, thereby significantly enhancing the efficacy of the verification process.

**Checklist Item Reweighting Analysis.** Here we present more reweighted case examples under a variety of tasks. As shown in Figure 14, the checklist generated for Coding & Debugging tasks ensures the complete generation of the code for the target problem. After reweighting, there is a greater emphasis on critical steps within the code, such as the structures of the forward, backward, and loss function in a neural network. For Planning & Reasoning tasks, reweighting makes key reasoning items more prominent. For example, As shown in Figure 17, based on the symptoms described in the problem, the response needs to deduce diabetic ketoacidosis (DKA) to provide a correct and reasonable treatment in subsequent answers. For other Creative Tasks, see Figure 15, and for Information/Advice seeking tasks, see Figure 16. By reweighting checklist items, we can reasonably focus on critical steps or results across different types of tasks, ensuring effectiveness when using lightweight LLM as judges.

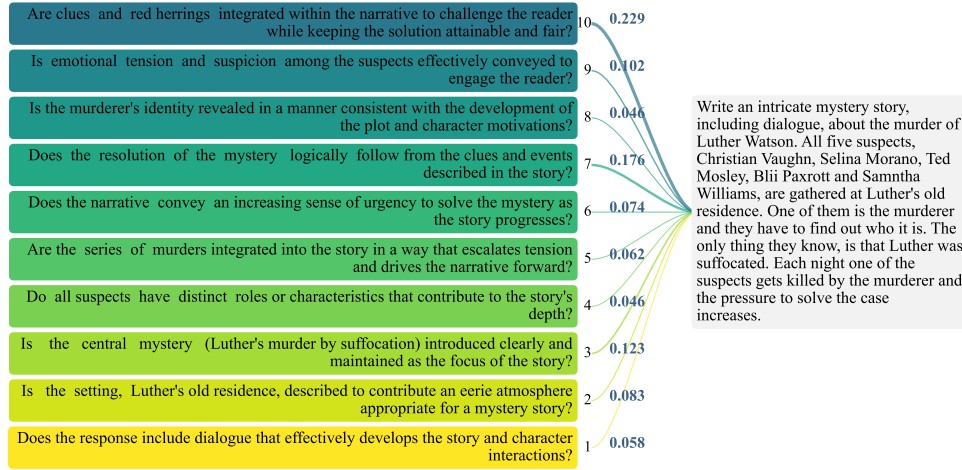

Figure 15: Visualization of checklist item reweighting in Creative Tasks.

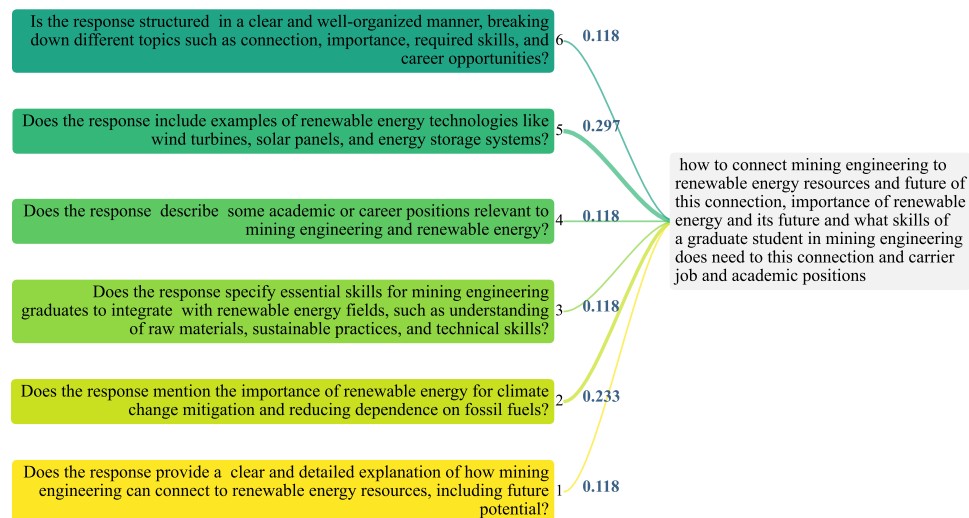

Figure 16: Visualization of checklist item reweighting in Information/Advice seeking.

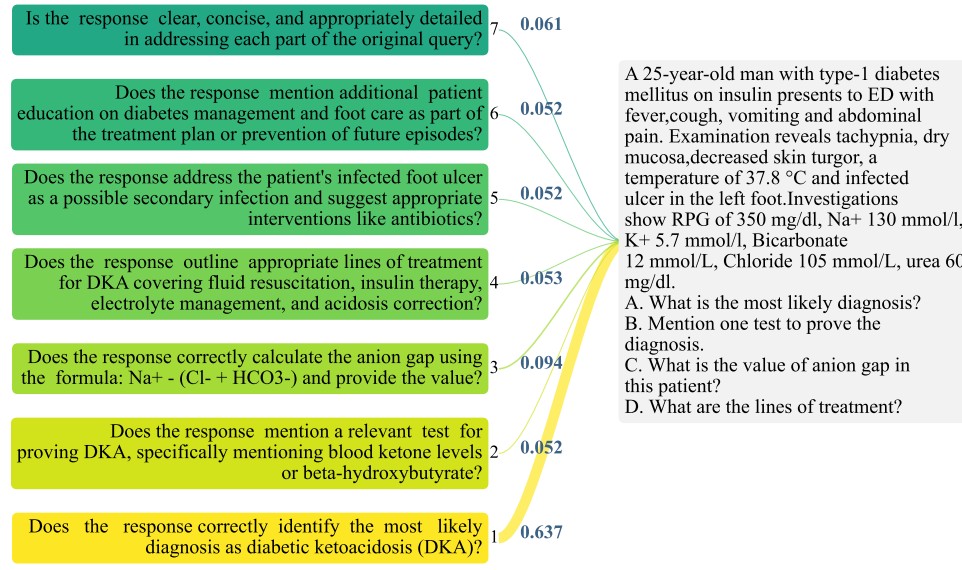

Figure 17: Visualization of checklist item reweighting in Planning & Reasoning.

