# OpenReview forum: "RocketEval: Efficient automated LLM evaluation via grading checklist"
_ICLR.cc/2025/Conference — ICLR 2025 Poster_

### Official Review · Reviewer_3Dkr · 2024-10-21

**Soundness:** 3
**Presentation:** 4
**Contribution:** 2
**Rating:** 6
**Confidence:** 4

**Summary:**

The authors propose RocketEval to address the high costs, privacy concerns, and reproducibility challenges associated with using powerful LLMs like GPT-4o for evaluation purposes. RocketEval leverages lightweight LLMs combined with a multi-faceted checklist approach to improve judgment accuracy. Experimental results from benchmarks like MT-BENCH and WILDBENCH demonstrate that RocketEval achieves a high correlation with human evaluations, comparable to GPT-4o, while reducing costs by more than 50-fold.

**Strengths:**

- I enjoyed reading the paper, especially Section 2, as the problem is well-motivated and the charts effectively convey the key messages.
- The idea is simple yet smart. By decomposing a single (potentially difficult) evaluation task into multiple simpler items in a checklist, lightweight LLMs can be leveraged to perform the evaluation. More importantly, checklist generation is largely a one-time effort, which could make LLM evaluation more economical in everyday LLM development environments.
- The authors have thoroughly evaluated the proposed method from multiple perspectives, including agreement, cost, and a qualitative study.

**Weaknesses:**

- Some aspects of the design lack scientific rigor. For instance, the estimation of “Normalized Probability” does not appear to be justified. Why would it provide an unbiased (or even plausible) probability? I suggest simply referring to it as a “score” instead of “probability”.
- Direct evaluation using lightweight LLMs is critiqued for high uncertainty and position bias. However, the authors fail to provide a comparison in this regard during the evaluation. Could this issue be addressed by bias mitigation algorithms [1, 2] at an even lower cost? What is the unique advantage offered by the checklist-based method compared to those approaches?
- The authors mentioned the potential benefits of using prefix caching for cost savings. However, this is not reflected in Section 4.2, even though this feature is already available through OpenAI.
- In Section 4.2, the authors quote the cost from `vast.ai`, a bidding-based crowd-sourced GPU rental platform. The quoted price ($0.8/hr) is the minimum price for renting an A100 card according to `https://vast.ai/pricing` (why not use the median price?). This might be unfair for several reasons. First, it is unlikely that $0.8/hr represents the regular price. More importantly, OpenAI offers production-level stability, which `vast.ai` does not. A fair comparison would be to quote the price from a public cloud provider like AWS.

[1] Large language models are not fair evaluators

[2] Split and merge: Aligning position biases in large language model based evaluators

**Questions:**

How does the method compare to other approaches, such as [1, 2], as well as fine-tuned judge LLMs?

---

> ### Author Response · Authors · 2024-11-19
>
> &emsp;We appreciate the reviewer's positive feedback. We are pleased that Section 2 was well-received for its clear motivation and effective charts. We also value the recognition of our method's simplicity and practicality in decomposing complex tasks, making LLM evaluations more economical. Additionally, the thorough multi-perspective evaluation of our method was acknowledged, enhancing its credibility. Below, we will address remained questions or concerns from the reviewer.
>
> > #### **W1: Some aspects of the design lack scientific rigor. For instance, the estimation of “Normalized Probability” does not appear to be justified. Why would it provide an unbiased (or even plausible) probability? I suggest simply referring to it as a “score” instead of “probability”.**
>
> Thank you for your valuable feedback. The term "probability" for the variable $p^(x,y,c)$ in Equation (1) was used based on the assumptions that:
>
> - LLMs produce a single token at once, making the output of "Yes" and "No" mutually exclusive events.
> - The LLM's output in this context is limited to "Yes" and "No," as requested in the prompt.
>
> However, we acknowledge that the second assumption is not strictly accurate in the LLM generation process without explicit hard masking of other tokens. To avoid potentially misleading terminology, we agree with your suggestion and have replaced all instances of "normalized probability" with "normalized score" in the revised version.
> - Revision: Section 3.2.
>
> > #### **W2: Direct evaluation using lightweight LLMs is critiqued for high uncertainty and position bias. However, the authors fail to provide a comparison in this regard during the evaluation. Could this issue be addressed by bias mitigation algorithms [1, 2] at an even lower cost? What is the unique advantage offered by the checklist-based method compared to those approaches?**
>
> - Our work demonstrates the impact of high uncertainty and position bias in lightweight LLMs. We address these issues with RocketEval, mitigating high uncertainty through independent checklist grading and position bias through normalized probability (now referred to as **normalized score**). The ablation study has been included in the appendix (Table 7 and Table 8) due to space constraints.
> - While algorithms [1] and [2] aim to address position bias, their focus is primarily on bias stemming from **response order in pairwise comparisons**. In contrast, our concern is the influence of prior judgments within a *single* lightweight LLM's analysis, where earlier judgments affect subsequent ones within the generated analysis itself.
>     - The first method in [1] utilizes CoT to generate analysis, does not address this specific type of position bias (**within a single analysis generation**) we focused on.
>     - The second method is swapping the position of two reponses, which is **impractical with a larger number of checklist questions**.
>     - [2] focuses on mitigating response order effects by segmenting semantically similar modules **between two responses**. Our work evaluates a **single response**, rendering this approach inapplicable.
>
> Therefore, while valuable, the bias mitigation strategies discussed in [1] and [2] are **not directly suitable** for addressing the specific type of position bias we observe.
> > #### **W3: The authors mentioned the potential benefits of using prefix caching for cost savings. However, this is not reflected in Section 4.2, even though this feature is already available through OpenAI.**
>
> - Our method benefits significantly from prefix caching because the inputs for the multiple checklist questions pertaining to a single query-response pair share nearly identical prefixes, differing only in the final checklist question.
> - Chain-of-Thought (CoT) based methods, which perform only a single inference per query-response pair, would only see substantial cost savings from prefix caching when there is a significant disparity between the query and response lengths.
> - While we acknowledge the recent introduction of OpenAI's [prompt caching feature](https://openai.com/api/pricing/), it is not currently applicable to batch inference. Therefore, the pricing we reference remains the lowest currently available.
>
>
> To better illustrate the cost benefits of prefix caching for our method, we compare the single-inference times with and without prefix caching on different LLM sizes. This result is summarized here for your convenience.
>
> | Model        | Prefix Caching Enabled | Prefix Caching Disabled |
> | ------------ | ---------------------- | ----------------------- |
> | Llama-3-8B   | 289s                   | 1406s                   |
> | Gemma-2-2B   | 157s                   | 446s                    |
> | Qwen2.5-1.5B | 110s                   | 307s                    |
>
> Since prefix caching is a commonly implemented feature in the LLM inference framework, we consider it as the default setting for cost estimation.

---

> ### Author Response · Authors · 2024-11-19
>
> > #### **W4: In Section 4.2, the authors quote the cost from vast.ai, a bidding-based crowd-sourced GPU rental platform. The quoted price ($0.8/hr) is the minimum price for renting an A100 card according to https://vast.ai/pricing (why not use the median price?). This might be unfair for several reasons. First, it is unlikely that $0.8/hr represents the regular price. More importantly, OpenAI offers production-level stability, which vast.ai does not. A fair comparison would be to quote the price from a public cloud provider like AWS.**
>
> - We agree that using *vast.ai* as a reference for cost estimation is unreliable due to its crowd-sourced, bidding-based nature and lack of production-level stability.
> - Furthermore, our experiments indicate that lightweight LLMs achieve more optimal inference efficiency on **lower-spec GPUs**. Therefore, we have revised this section as follows:
>
> We re-evaluated the cost of models smaller than 8B parameters using an **RTX A6000**.  We referenced pricing from multiple cloud providers, as detailed in the following table:
> | Provider                     | A6000 Price (per GPU / hour) | A100 Price (per GPU / hour) |
> |------------------------------|-----------------------|------------------------------|
> | [Cudo Compute](https://www.cudocompute.com/pricing) | \$0.45                   | \$1.50 |
> | [TensorDock](https://dashboard.tensordock.com/deploy?gpu=rtxa6000-pcie-48gb) | \$0.40                   | $1.62 |
> | [Runpod](https://www.runpod.io/pricing) (Community) | \$0.49 | \$1.19 |
> | [Runpod](https://www.runpod.io/pricing) (Secure) | \$0.76 | \$1.64 |
> | [HyperStack](https://www.hyperstack.cloud/gpu-pricing) | \$0.50                   | \$1.35 |
> | [Oblivus Cloud](https://oblivus.com/pricing)      | \$0.55                   | \$1.47 |
> | [Massed Compute](https://massedcompute.com/home/pricing/) | \$0.625                  | \$1.72 |
> | [AWS](https://calculator.aws/#/) | N/A                  | \$1.98 (3 year plan) |
>
> We ultimately selected [RunPod](https://www.runpod.io/) Secure Cloud's pricing (A100: \\$1.64/hour, A6000: \\$0.76/hour) for calculations. These updated results are presented in Table 4.  As demonstrated, despite using a more reliable and potentially higher cost basis, leveraging lower-spec GPUs did not negatively impact inference efficiency on lightweight LLMs, and, in fact, resulted in an **overall cost reduction.**
>
> - Revision: Table 4, L467.
>
> > #### **Q1: How does the method compare to other approaches, such as [1, 2], as well as fine-tuned judge LLMs?**
>
> - For the difference between our approach and [1] and [2]，please refer to the previous response (Response to W2). We believe that they focus on alleviating position bias in **pairwise comparison**, which is different from our approach that focuses on **point-wise evaluation**.
> - For fine-tuned LLM judge model, we have incorporated the latest model Prometheus-v2 [3], as a baseline in our evaluation. The results of this comparison are presented in Tables 2 and 3. Results show that the Gemma-2-2B model with RocketEval can outperform the 7B fine-tuned judge model.
>
> ------
>
> We hope that our responses have adequately addressed the reviewers' questions and concerns. We are more than willing to discuss any further questions the reviewer may have.
>
> ### **Reference**
> [1] Large language models are not fair evaluators. ACL '24.
>
> [2] Split and merge: Aligning position biases in large language model based evaluators. EMNLP '24.
>
> [3] Prometheus 2: An Open Source Language Model Specialized in Evaluating Other Language Models. arXiv.

---

> > ### Comment · Reviewer_3Dkr · 2024-11-19
> >
> > Thank you for your clarification. Most of my concerns have been addressed. However, I noticed that many of the benchmarks you used involve pairwise comparisons (cf. Table 5). Could you elaborate on why a direct comparison with [1] and [2] is not feasible in this context, or am I misunderstanding something?

---

> ### Author Response · Authors · 2024-11-20
>
> Thank you for your reply.
>
> The primary reason for not directly comparing our method with [1,2] is the difference in the **types of position bias** addressed and the different **types of evaluation methods** adopted.
> - Existing work on LLMs as judges can be categorized into point-wise methods, which score **individual responses** (**our focus**), and pairwise methods, which **compare two responses** to select a better one (**the focus of [1,2]**).
> - The position bias explored in [1,2] refers to the influence of the **order in which two responses are presented**. In our point-wise setting, we only input a single response at a time, **eliminating this specific type of bias**.
> - Therefore, our paper focuses on a **different concept of position bias** than [1,2], and our **evaluation methodology differs** accordingly (pairwise vs. point-wise). While a direct comparison could be performed, we believe it would **not yield meaningful insights**.
> - Consequently, we focused our comparisons on methods employing the **same point-wise evaluation**, including those using CoT generated analysis and fine-tuned judge LLMs.
>
> Furthermore, based on your feedback, we recognize the potentia misleading information in Table 5. The "point-wise" and "pairwise" labels in Table 5 actually refer to the evaluation methods used by the benchmark creators [3-6], not a restriction on the applicability of the datasets themselves. **Any dataset could be used for either point-wise or pairwise evaluation method.** In our experimental setting, all experiments are conducted in a **point-wise** manner.
>
> In consideration of your feedback, we will make changes in the next revision:
> - Clarify the distinctions between our work and [1,2] in our discussion of existing works.
> - Remove the potentially misleading information from Table 5.
>
> Thank you again for your valuable feedback. We hope this clarification addresses your concerns.
>
> ### **Reference**
> [1] Large language models are not fair evaluators. ACL '24.
>
> [2] Split and merge: Aligning position biases in large language model based evaluators. EMNLP '24.
>
> [3] Judging LLM-as-a-Judge with MT-Bench and Chatbot Arena. NIPS '23.
>
> [4] WildBench: Benchmarking LLMs with Challenging Tasks from Real Users in the Wild. arXiv.
>
> [5] AlpacaFarm: A Simulation Framework for Methods that Learn from Human Feedback. NIPS '23.
>
> [6] From crowdsourced data to high-quality benchmarks: Arena-hard and benchbuilder pipeline. arXiv.

---

> > ### Comment · Reviewer_3Dkr · 2024-11-21
> >
> > Thank you for your clarification. Your explanation addresses my concerns. Based on this, I am raising my score to 6.

---

### Official Review · Reviewer_Pg23 · 2024-10-28

**Soundness:** 3
**Presentation:** 3
**Contribution:** 2
**Rating:** 6
**Confidence:** 4

**Summary:**

This paper introduces RocketEval, a lightweight and automated evaluation method for LLMs that addresses the high costs and privacy concerns of using powerful LLMs as judges. By reframing evaluation tasks into a Q&A format using instance-specific checklists, RocketEval improves the judgment accuracy of smaller models, overcoming issues like uncertainty and positional bias.

**Strengths:**

- Clear Motivation: The author clearly explains the motivation of the paper in the introduction. I particularly agree with the point raised about the shortcomings of "Fine-Tuned Judge Models," as these models often lack understanding of complex instructions (although the author did not provide proof for this claim).
- Overall Well-Written: The paper is generally well-written.
- Relevant Problem: The paper raises a highly relevant problem that urgently needs to be addressed in the current landscape of LLM evaluation.

**Weaknesses:**

- Regarding Section 2.4: While information entropy is a good metric, it is not very intuitive. I recommend that the author use self-consistency (sampling n times and calculating consistency) to demonstrate this uncertainty.
- Limited Advantage Over GPT-4o-Mini: Although RocketEval shows a clear advantage over GPT-4o, its superiority over GPT-4o-mini is less significant. Additionally, the N=1000 setting is impractical, as it is rare for someone to test 1000 models simultaneously.
- Missing Ablation Studies: The paper lacks ablation studies for two components: Independent Checklist Item Judgment and Normalized Probability.
- Comparison with Other Models: I also suggest that the author provide comparisons with other Fine-Tuned Judge Models, such as the Prometheus series.
- Minor Issues: In lines 289-290, the author mentions that reference responses and hand-crafted rubrics may not work well in all cases. Please provide examples to illustrate this. Figure 10 is confusing—why does the first row have three models, while the following subplots have only one model per row?

Overall, I believe the problem and methods proposed in the paper are valuable, but there is room for improvement. I hope the author can address my concerns during the rebuttal stage.

**Questions:**

Please see the weaknesses.

---

> ### Author Response · Authors · 2024-11-19
>
> &emsp;We appreciate the reviewer's positive feedback and constructive comments. We are pleased to learn that the motivation of our paper was clearly articulated and resonated with the reviewer, especially regarding the critique of "Fine-Tuned Judge Models." Additionally, we are glad that the paper's relevance to current issues in LLM evaluation was recognized and that the overall writing quality was commended. Below, we will address remained questions or concerns from the reviewer.
>
> > #### **W1: Regarding Section 2.4: While information entropy is a good metric, it is not very intuitive. I recommend that the author use self-consistency (sampling n times and calculating consistency) to demonstrate this uncertainty.**
>
> - Thank you for this valuable suggestion. To enhance clarity, we have redesigned the task in Section 2.4 as you suggested. We now demonstrate the uncertainty of lightweight LLMs by comparing **the ratio of disagreement in N sampling results** among all checklist questions.
> - This revised approach also offers the benefit of **aligning the experimental design** for demonstrating uncertainty with that used for validating position bias, thereby improving the overall coherence and ease of understanding for the reader.
> - Revisions: Figure 4, L237-241.
>
> > #### **W2: Limited Advantage Over GPT-4o-Mini: Although RocketEval shows a clear advantage over GPT-4o, its superiority over GPT-4o-mini is less significant. Additionally, the N=1000 setting is impractical, as it is rare for someone to test 1000 models simultaneously.**
>
> - Here the value of N=1000 does **not** refer to the number of **distinct models**. In LLM development, variations in prompting techniques and sampling strategies can lead to numerous different outputs for testing. Furthermore, it's important to clarify that these 1000 evaluations are **not** necessarily performed **at once**. Previous results can be reused for comparison when the dataset remains unchanged. Therefore, the cost can be considered **cumulative over time**.
> - For these reasons, we maintain that N=1000 is a reasonable and representative choice for showcasing the cost-effectiveness of our method. To reflect this clarification, we have modified Table 4 (from "N=1000 models" to "N=1000 **tests**").
> - At the same time, although RocketEval method does not generate dozens of times of cost savings compared to GPT-4o-mini, the efficiency improvement it brings is still **highly considerable**. In addition, in some **offline and privacy-concerned** evaluation scenarios, RocketEval has its value in providing efficient and human preference aligned evaluation based on a lightweight open source LLM, a capability not readily available with closed-source alternatives.
> - Revision: Table 4.
>
> > #### **W3: Missing Ablation Studies: The paper lacks ablation studies for two components: Independent Checklist Item Judgment and Normalized Probability.**
>
> - We have already included an ablation study for normalized probability (now referred to as **normalized score**) in the appendix (*Table 7 & 8*).
> - We have also **added an ablation study** for independent checklist item judgment (Indep. Judgment) with further discussions in the revised version, as presented in Tables 7 & 8.
> -  Revisions: Table 7 & 8, L1261-1267.

---

> ### Author Response · Authors · 2024-11-19
>
> > #### **W4: Comparison with Other Models: I also suggest that the author provide comparisons with other Fine-Tuned Judge Models, such as the Prometheus series.**
>
> - We have already incorporated the latest fine-tuned LLM judge model, **Prometheus-v2** [1], as a baseline in our evaluation. The results of this comparison are presented in Tables 2 and 3. Results show that the Gemma-2-2B model with RocketEval can outperform the 7B fine-tuned judge model.
>
> > #### **W5-1: Minor Issues: In lines 289-290, the author mentions that reference responses and hand-crafted rubrics may not work well in all cases. Please provide examples to illustrate this.**
>
> Here, our original intent was to **highlight the limitations** of both reference responses and hand-crafted rubrics.
> - Reference responses are often inadequate for evaluating **open-ended questions** due to the diversity and complexity of potential valid answers.
> - Hand-crafted rubrics, while potentially more comprehensive, face a trade-off between cost, generalizability, and accuracy. [2] illustrates this trade-off by employing two modes of rubric utilization.
>     - For fine-tuning, they leverage an augmented rubric set to generate instruction data, ensuring alignment between the rubric and the data. However, this approach is not applicable to **pre-existing datasets**.
>     - For evaluation, they use both **generic rubrics** (e.g., helpfulness, harmlessness) and **case-by-case hand-crafted rubrics**. While generic rubrics offer broader applicability, they provide **limited guidance** for analyzing nuanced responses. Conversely, case-by-case rubrics, while precise, **incur high development costs**, hindering scalability.
>
> Given these limitations, we believe that automated checklist creation offers a more **viable and scalable solution** for LLM evaluation.
>
> We agree that the original phrasing was too general and lacked sufficient explanation. We have therefore revised this section to provide more specific examples and clarify our reasoning for pursuing automated checklist generation in RocketEval.
> - Revision: L283-288.
>
> > #### **W5-2: Figure 10 is confusing—why does the first row have three models, while the following subplots have only one model per row?**
>
> - In the original Figure 10, the first row displayed results from three powerful models (GPT-4o, Llama-3-70B, and Qwen-2-72B) using CoT as baselines. We placed them in a single row **to conserve space**.
> - However, we acknowledge that this presentation was confusing. We have revised Figure 10 to **only include the results from GPT-4o** as a baseline, streamlining the visualization and facilitating a clearer comparison with the lightweight LLMs presented in the subsequent rows.
> - Revision: Figure 10.
> ------
>
> We hope that our responses have adequately addressed the reviewers' questions and concerns. We are more than willing to discuss any further questions the reviewer may have.
>
> ### **Reference**
> [1] Prometheus 2: An Open Source Language Model Specialized in Evaluating Other Language Models. arXiv.
>
> [2] Prometheus: Inducing Fine-grained Evaluation Capability in Language Models. ICLR '24.

---

> > ### Comment · Reviewer_Pg23 · 2024-11-25
> >
> > Thanks for your response. Most concerns are addressed and I have raised my score. Good luck.

---

### Official Review · Reviewer_3Tvf · 2024-10-29

**Soundness:** 3
**Presentation:** 4
**Contribution:** 3
**Rating:** 6
**Confidence:** 4

**Summary:**

This paper proposes RocketEval, a novel framework for efficiently evaluating large language models (LLMs) using lightweight LLMs as judges. The framework addresses the limitations of existing automated evaluation methods, such as high costs and lack of interpretability, by using checklist grading. This paper trains lightweight LLMs to independently evaluate each checklist item, providing a multifaceted and unbiased judgment.

**Strengths:**

- Originality: The paper presents a novel evaluation framework, RocketEval, which evaluates LLMs by grading checklist. This approach is distinct from existing methods like multiple-choice questions. However, I believe the idea of using checklist to evaluate NLP models was actually proposed by another paper [1] in 2020.
- Quality: The analysis of the lightweight LLMs' abilities in section 2 provides valuable insights into the framework’s effectiveness.
- Clarity: The paper is well-organized and clearly structured, with each section logically following the previous one.
- Significance: The framework’s high agreement with human judgments and significant cost reduction make it a promising solution for large-scale LLM evaluations.


[1] Ribeiro, Marco Tulio, Tongshuang Wu, Carlos Guestrin, and Sameer Singh. "Beyond accuracy: Behavioral testing of NLP models with CheckList." arXiv preprint arXiv:2005.04118 (2020).

**Weaknesses:**

- The paper primarily focuses on evaluating responses to queries, as the checklist is generated by the input query. However, LLMs are used in a wide range of applications, including text summarization, and translation, where evaluation metrics based on queries may not be efficient since the input query may be very long.
- Since authors employ GPT-4o as the checklist creator, while the paper argues that checklist creation is a one-time process, the cost of using a powerful LLM like GPT-4o for this task could be significant, especially for large-scale evaluations. Exploring more efficient methods for checklist creation would be beneficial.
- Typo: Line 215: We select responses from 12 test models and compare the benchmark results when using *GPT-4o* and *Claude-3.5-Sonnet* as judges as strong baselines.

**Questions:**

- Could authors explain their framework's relation and difference with paper [1] ?
- RocketEval can enable lightweight models the ability to judge large models, is this similar to the idea of weak-to-strong supervision? Please explain the relation and difference between your paper's ideas and weak-to-strong [2].
- Conducting experiments solely on a total of 1104 data from MT-BENCH and WILDBENCH is not convincing enough. Could authors conduct experiments on more datasets?

[1] Ribeiro, Marco Tulio, Tongshuang Wu, Carlos Guestrin, and Sameer Singh. "Beyond accuracy: Behavioral testing of NLP models with CheckList." arXiv preprint arXiv:2005.04118 (2020).

[2] Burns, Collin, Pavel Izmailov, Jan Hendrik Kirchner, Bowen Baker, Leo Gao, Leopold Aschenbrenner, Yining Chen et al. "Weak-to-strong generalization: Eliciting strong capabilities with weak supervision." arXiv preprint arXiv:2312.09390 (2023).

---

> ### Author Response · Authors · 2024-11-19
>
> &emsp;We thank the reviewer for their constructive feedback. We are encouraged by the recognition of the originality of our RocketEval framework. We also appreciate the positive remarks on the clarity and logical structure of our paper, as well as the framework's effectiveness in aligning closely with human judgments and reducing evaluation costs. Below, we will address remained questions or concerns from the reviewer.
>
>
> > #### **W1: The paper primarily focuses on evaluating responses to queries, as the checklist is generated by the input query. However, LLMs are used in a wide range of applications, including text summarization, and translation, where evaluation metrics based on queries may not be efficient since the input query may be very long.**
>
> - Since our method is designed for general LLM evaluation scenarios, we do not target at **controlling the input length** as a means to improve efficiency.
> - For specific tasks such as text summarization and translation, especially with long input queries, reference-based metrics like BLEU and ROUGE are considered as applicable and effcient evaluation metrics. However, existing studies have questioned the **generalizability** of these metrics and their **relevance to human judgment**, and results suggest that approaches that have large language models (LLMs) act as judges are superior [1,2,3].
> - In the topic of **LLM-as-a-judge**, by employing a more lightweight LLM for human preference-aligned evaluation, our method remains more efficient compared to other LLM-as-judge approaches on such tasks.
>
>
> > #### **W2: Since authors employ GPT-4o as the checklist creator, while the paper argues that checklist creation is a one-time process, the cost of using a powerful LLM like GPT-4o for this task could be significant, especially for large-scale evaluations. Exploring more efficient methods for checklist creation would be beneficial.**
>
> Thank you for this valuable suggestion. We acknowledge that utilizing GPT-4o to generate the checklist incurs a relative high cost. However, in large-scale LLM evaluation scenarios where numerous models, prompting techniques and sampling parameters result in a **massive volume of outputs for testing**, the computational cost of the checklist grading process dominates, as illustrated in Table 4. Therefore, the one-time cost of checklist creation becomes relatively less significant and acceptable.
>
> Nevertheless, we agree that exploring alternative, more efficient methods for checklist creation is beneficial for broader applicability and potential improvements in checklist quality. We will investigate this avenue further in our future work.
>
> > #### **W3: Typo: L215: We select responses from 12 test models and compare the benchmark results when using GPT-4o and Claude-3.5-Sonnet as judges as strong baselines.**
>
> Typo fixed in revision: "*Then, the scores of all responses are averaged to get the average score of the tested model for ranking. **For baselines, we derive the score predicted by GPT-4o and Claude-3.5-Sonnet as judges with CoT equipped.** ... Figure 3 shows the scores of selected **12 test models** and the Spearman correlation coefficient of ranking lists with GPT-4o when different LLMs served as judge.*"
> - Revision: L215-216.
>
> > #### **Q1: Could authors explain their framework's relation and difference with paper [4] ?**
>
> - The paper [4] proposes a task-agnostic method called "CHECKLIST". Although both CHECKLIST and Rocketeval are designed for evaluating language models, they have many differences：
>
> | Feature | RocketEval | CHECKLIST [4] |
> |---|---|---|
> | Scope | **Universal** LLM evaluation applicable to various tasks, including creative generation. | Focuses on **identifying critical failures** in language model. Limited in evaluating specific creative generation tasks. |
> | Automation | Fully **automated** evaluation process. | Requires **human involvement** in formulating rules, customizing templates, and adapting to new domains. |
> | Data | **Independent** of test dataset. | Rely on creation of test cases from scratch or perturbations of existing datasets following the **task-specific** abstract template. |
>
> - In summary, we believe that RocketEval is designed as an efficiency improvement for LLM-as-a-judge, which is significantly different from the objectives of CHECKLIST: **identifying the critical failures of language models.**

---

> ### Author Response · Authors · 2024-11-19
>
> > #### **Q2: RocketEval can enable lightweight models the ability to judge large models, is this similar to the idea of weak-to-strong supervision? Please explain the relation and difference between your paper's ideas and weak-to-strong [5].**
>
> Thank you for reading the papers in detail and noticing the subtle similarities between [5] and RocketEval. We bellieve the core distinction lies in the nature of the tasks.
> - In [5], both weak and strong models are evaluated on the **same task**, with the weak model providing supervision signals for fine-tuning of the strong model.
> - In the LLM-as-a-Judge scenario, without additional information, we hypothesize that a judge model should possess stronger capabilities than the tested model to judge its responses correctly, **similar to the setting** in [5].
> - However, RocketEval introduces a crucial difference: **the lightweight LLM judge does not need to independently understand and complete the task presented in the original query**. Instead, it performs binary judgments on a checklist decomposed by a stronger LLM. This creates a clear distinction between the task of the lightweight LLM judge and the task of the model being evaluated.
> - Furthermore, the strong model provides additional information during checklist creation, meaning the **supervision signal during evaluation isn't solely from the lightweight LLM judge**. This is a significant departure from traditional weak-to-strong supervision, where **source of supervision only comes from weak model**.
>
> Therefore, while the concept of utilizing a less powerful model has similarities to weak-to-strong supervision, the distinct task structure and the role of the stronger LLM in checklist generation differentiate RocketEval from the approach in [5].
>
>
>
> > #### **Q3: Conducting experiments solely on a total of 1104 data from MT-BENCH and WILDBENCH is not convincing enough. Could authors conduct experiments on more datasets?**
>
> In addition to MT-Bench and WildBench, we have also conducted experiments on **AlpacaEval** [6] and **Arena-Hard** [7]. These datasets encompass diverse task types and exhibit variations in length and number of turns.
>
> Due to space limitations, these results are presented in the appendix (*Table 6*). The results consistently demonstrate the effectiveness of our method in improving the evaluation capabilities of lightweight LLM judges **across all datasets**.
>
> ------
>
> We hope that our responses have adequately addressed the reviewers' questions and concerns. We are more than willing to discuss any further questions the reviewer may have.
>
> ### **Reference**
> [1] G-EVAL: NLG Evaluation using GPT-4 with Better Human Alignment. EMNLP '23.
>
> [2] Bartscore: Evaluating generated text as text generation. NIPS '21.
>
> [3] Hurdles to progress in long-form question answering. NAACL '21.
>
> [4] Beyond Accuracy: Behavioral Testing of NLP models with CheckList. ACL '20.
>
> [5] Weak-to-Strong Generalization: Eliciting Strong Capabilities With Weak Supervision. arXiv.
>
> [6] Alpacafarm: A simulation framework for methods that learn from human feedback. NIPS '23.
>
> [7] From crowdsourced data to high-quality benchmarks: Arena-hard and benchbuilder pipeline. arXiv.

---

> ### Comment · Reviewer_3Tvf · 2024-11-19
> **Thank you**
>
> Thanks for your responses. I have read the rebuttal and I think an overall rating of 6 is reasonable.

---

### Official Review · Reviewer_TQvF · 2024-10-30

**Soundness:** 2
**Presentation:** 3
**Contribution:** 2
**Rating:** 6
**Confidence:** 3

**Summary:**

The authors propose RocketEval, a system comprised of API calls and lightweight models, performing as a model judge. The system is designed to be minimally computationally expensive while achieving comparable human agreement scores to GPT4o on MT-Bench and WildBench. Their subsequent ablations and analysis inform how the system can achieve these efficiency gains.

**Strengths:**

* The authors core objective is important and compellingly achieved: create a lightweight model judge with high performance. Their use of a dynamic checklist, and normalized score predictions both show notable improvements in scoring, from their experiments.
* The authors unpack the issues with existing lightweight model judges, related to their analyses, position bias and uncertainty, that could be informative to other solutions as well.
* The comprehensively compare a series of model judges in different settings to understand how they can best be optimized. Their quantitative analysis and ablations are particularly informative.

**Weaknesses:**

* Their explanation for lightweight model “analysis” being weak could be framed more conservatively. If I understand correctly, these experiments more show that lightweight models find classifying GPT4o reviews into scores as an easier task than predicting the score themselves. It wouldn’t necessarily naturally follow that a GPT4o checklist would be the obvious solution, but rather a GPT4o reasoning.
* It isn’t immediately apparent why GPT4o is so much more expensive than the RocketEval (which includes GPT4o calls). This should be more closely explained. Is it because the reasoning is super long? If so, it would be beneficial to see harder baselines (in terms of efficiency), e.g. prompting GPT4o to produce shorter CoT explanations before its score prediction.

**Questions:**

In Table 3, human-human agreement seems very low. Does this saturate the performances above 60% to some extent?

---

> ### Author Response · Authors · 2024-11-19
>
> &emsp;We thank the reviewer for their positive feedback. We are pleased that our core objective of creating a high-performing, lightweight model judge was recognized as important and effectively achieved. The reviewer's appreciation for our innovative methods, such as the dynamic checklist and normalized score predictions, and our comprehensive comparative analysis of model judges, is highly encouraging. Below, we will address remained questions or concerns from the reviewer.
>
> > #### **W1: Their explanation for lightweight model “analysis” being weak could be framed more conservatively. If I understand correctly, these experiments more show that lightweight models find classifying GPT4o reviews into scores as an easier task than predicting the score themselves. It wouldn’t necessarily naturally follow that a GPT4o checklist would be the obvious solution, but rather a GPT4o reasoning.**
>
> - In Section 2.3 of our paper, we conducted experiment to **demonstrate the limitations** of lightweight LLMs in analyzing. In Section 2.4, by breaking down the reasoning process of analyzing into judgments of a series of checklist questions, we further showed that this limitation might be caused by high uncertainty and position bias. This subsequently led to the introduction of a checklist grading method as a potential solution.
> - We agree that the conclusion of Section 2.3 **does not naturally lead to the motivation for a checklist-based solution**, which is primarily introduced in the discussion of Section 2.4. To address this clarity issue, we have made revisions to the conclusion of Section 2.3 and the introduction of Section 2.4 accordingly, ensuring a more cohesive transition and rationale for our proposed method.
> - Revision: L230-233.
>
> > #### **W2: It isn’t immediately apparent why GPT4o is so much more expensive than the RocketEval (which includes GPT4o calls). This should be more closely explained. Is it because the reasoning is super long? If so, it would be beneficial to see harder baselines (in terms of efficiency), e.g. prompting GPT4o to produce shorter CoT explanations before its score prediction.**
>
> - Although our method employs GPT-4o to generate a Checklist, it only needs to be executed once per query, after which it can evaluate an arbitrary number of corresponding responses. Therefore, this approach can result in cost savings, especially in the LLM evaluation scenario where different models, prompting techniques and sampling strategies can lead to **numerous number of tests**.
> - Based on the data collected from [Wildbench](https://github.com/allenai/WildBench/tree/main/eval_results/v2.0625/score.v2/eval%3Dgpt-4o-2024-05-13), the output length when using GPT-4o for evaluation is approximately **200 tokens** (228k tokens for 1k queries), which is considered to be within a reasonable range.
> - In addition, referring to Table 4, it is notable that while the output tokens are more expensive in GPT-4o, the cost of input tokens still accounts for more than 50% of the total cost. Therefore, our conclusions remain unaffected even when output cost is not considered.
>
> > #### **Q1: In Table 3, human-human agreement seems very low. Does this saturate the performances above 60% to some extent?**
>
> - The human-human agreement data can be refer to MT-Bench [1]. Given that MT-Bench includes a diverse range of tasks, including both closed-ended and open-ended, we consider a 64.8% human-human agreement to be a **reasonable** value.
> - Also, we agree with your observation that this metric might indeed saturate around 60%. However, this does **not contradict the objective** of our paper: align **lightweight LLM** with human preferences to improve the efficiency of LLM evaluation.
> - Our approach successfully elevates the agreement level of a lightweight Qwen2.5-1.5B model from a **low-level** (46.5%) to **near saturation** (60.7%). This marks a significant improvement in how well the lightweight LLM aligns with human preferences when RocketEval is adopted.
>
> ------
> We hope that our responses have adequately addressed the reviewers' questions and concerns. We are more than willing to discuss any further questions the reviewer may have.
>
> ### **Reference**
>
> [1] Judging LLM-as-a-Judge with MT-Bench and Chatbot Arena. NIPS '23.

---

> > ### Comment · Reviewer_TQvF · 2024-11-25
> >
> > Thank you for the clarifications. I will raise my score to a 7.

---

### Author Response · Authors · 2024-11-19

&emsp;First of all, we would like to thank the reviewers for recognizing and supporting our work and for their many thoughtful suggestions on how to improve the presentation of the paper. We are pleased to hear the positive feedback from reviewers, including clear motivation (*Pg23,3Dkr*), well presentation (*3Tvf,Pg23*), extensive experiments (*TQvF,3Dkr*), and the significance of the value to LLM research (*3Tvf,Pg23,3Dkr*). Meanwhile, based on the concerns and comments raised by multiple reviewers, we have revised the paper. Here is a comprehensive overview of the changes in the revision.

**Supplemental Results on Ablation Study (Appendix A.3, Table 7 & 8)**: Addressing feedback from Reviewer *Pg23*, we have added an ablation study focusing on independent checklist item judgment with further discussion.

**Updated References of Cost Estimation (Sec. 4.2, Table 4)**: In response to Reviewer *3Dkr*, we have recalculated costs using data from more reliable GPU cloud service providers. This adjustment ensures a fairer comparison to demonstrate the high efficiency of RocketEval.

**Improved Experiment Design (Sec. 2.4, Fig. 4)**: As suggested by Reviewer *Pg23*, we have redesigned the experiment in Section 2.4. We now illustrate the high uncertainty of lightweight LLMs using the ratio of disagreement in multiple sampling results instead of information entropy. This approach offers clearer insights and aligns with subsequent experiments for improved coherence.

**Revised Terminology (Sec. 3.2)**: Following Reviewer *3Dkr*'s suggestion, we have replaced "normalized probability" with "normalized score" in the introduction of RocketEval, aiming to increase rigor and avoids potential misunderstandings.

In addition, based on feedbacks from all reviewers, we have refined several sections of the manuscript to enhance clarity and consistency throughout the paper.
- **Clarified Statements and Descriptions: L230-232, L284-288.**
- **Revised Typos: L215-216, Table 4.**

---

> ### Author Response · Authors · 2024-11-23
>
> We thank the reviewers for helping us improve the quality of our paper. Based on the [feedback](https://openreview.net/forum?id=zJjzNj6QUe&noteId=0Ojo7aaNe1) from reviewer *3Dkr*, we have made the following revisions to the paper.
>
> - We have added the discussion and comparison on the position bias discussed in different studies: **L254-L257**.
> - Removal of misleading information (**Table 5**): The "evaluation" labels (pairwise or point-wise) in the original Table 5 actually refer to **the evaluation methods used by the benchmark creators**,  rather than characteristics of the datasets themselves. We have removed this information in the revision to avoid misleading information.

---

### Comment · Area_Chair_ksDn · 2024-11-25

Dear reviewers,

As the deadline for discussion is ending soon. Please respond to the authors to indicate you have read their rebuttal. If you have more questions, now is the time to ask.

AC

---

### Meta-Review · Area_Chair_ksDn · 2024-12-18

**Metareview:**

The paper introduces RocketEval, a framework for efficient LLM evaluation using lightweight models as judges. The system addresses critical challenges, including high computational costs, privacy concerns, and limited interpretability. By employing a dynamic checklist approach and leveraging API calls with lightweight models, RocketEval demonstrates remarkable performance, achieving high correlation with human evaluations while significantly reducing evaluation costs. Key strengths include its interesting decomposition of evaluation tasks, comprehensive analysis of lightweight model capabilities, and ability to provide multifaceted judgments across different benchmarks like MT-Bench and WildBench. Reviewers particularly appreciated the paper's clear motivation, well-structured presentation, and the potential for making large-scale LLM evaluations more accessible and cost-effective (3Tvf, Pg23, 3Dkr).

Despite its promising approach, the paper faces several weaknesses. Reviewers highlighted concerns about the its scientific rigor, including the lack of thorough ablation studies and unclear justification for certain components like "Normalized Probability" (3Dkr, Pg23). The reliance on GPT-4o for checklist creation raises questions about the actual cost-effectiveness, and the experimental scope appears limited, with experiments conducted on a relatively small dataset (3Tvf, Pg23). Additionally, the paper lacks comprehensive comparisons with alternative evaluation methods, such as fine-tuned judge models or bias mitigation algorithms (3Dkr, Pg23). Reviewers also noted potential improvements, including exploring more efficient checklist creation methods, conducting experiments on broader datasets, and providing clearer explanations of the method's novelty compared to existing approaches (3Tvf, Pg23). These limitations, while significant, did not overshadow the paper's potential contributions, with reviewers ultimately rating it marginally above the acceptance threshold.

Overall, this paper can be accepted as a poster. But the authors should address the concerns of the reviewers and reflect them in the final version.

**Additional Comments On Reviewer Discussion:**

All concerns have been addressed well and all reviewers agreed to increase the score to 6.

---

### Decision · Program_Chairs · 2025-01-22

Accept (Poster)